# Arabidopsis ACINUS is O-glycosylated and regulates transcription and alternative splicing of regulators of reproductive transitions

Yang Bi[1,2], Zhiping Deng [1], Weimin Ni[3], Ruben Shrestha[1], Dasha Savage[1], Thomas Hartwig[1], Sunita Patil[1], Su Hyun Hong[1], Zhenzhen Zhang[1], Juan A. Oses-Prieto [4], Kathy H. Li[4], Peter H. Quail[3], Alma L. Burlingame [4], Shou-Ling Xu [1✉] & Zhi-Yong Wang [1✉]

O-GlcNAc modification plays important roles in metabolic regulation of cellular status. Two homologs of O-GlcNAc transferase, SECRET AGENT (SEC) and SPINDLY (SPY), which have O-GlcNAc and O-fucosyl transferase activities, respectively, are essential in *Arabidopsis* but have largely unknown cellular targets. Here we show that AtACINUS is O-GlcNAcylated and O-fucosylated and mediates regulation of transcription, alternative splicing (AS), and developmental transitions. Knocking-out both AtACINUS and its distant paralog AtPININ causes severe growth defects including dwarfism, delayed seed germination and flowering, and abscisic acid (ABA) hypersensitivity. Transcriptomic and protein-DNA/RNA interaction analyses demonstrate that AtACINUS represses transcription of the flowering repressor *FLC* and mediates AS of *ABH1* and *HAB1*, two negative regulators of ABA signaling. Proteomic analyses show AtACINUS's O-GlcNAcylation, O-fucosylation, and association with splicing factors, chromatin remodelers, and transcriptional regulators. Some AtACINUS/AtPININ-dependent AS events are altered in the *sec* and *spy* mutants, demonstrating a function of O-glycosylation in regulating alternative RNA splicing.

[1] Department of Plant Biology, Carnegie Institution for Science, Stanford, CA, USA. [2] Department of Biology, Stanford University, Stanford, CA 94305, USA. [3] Plant Gene Expression Center, United States Department of Agriculture/Agriculture Research Service, Albany, CA, USA. [4] Department of Pharmaceutical Chemistry, University of California, San Francisco, San Francisco, CA, USA. ✉email: slxu@stanford.edu; zywang24@stanford.edu

Posttranslational modification (PTM) of intracellular proteins by O-linked *N*-acetylglucosamine (O-GlcNAc) is an important regulatory PTM that modulates protein activities and thereby controls cellular functions according to nutrient and energy status[1,2]. Extensive studies in animals have shown that thousands of proteins involved in diverse biological processes are modified on serine and threonine residues by O-GlcNAcylation, which is catalyzed by O-GlcNAc transferase (OGT) using UDP-GlcNAc as donor substrate[1,3]. As a sensor of primary metabolic status, O-GlcNAcylation plays key roles in cellular homeostasis and responses to nutritional and stress factors[1,2,4,5], whereas dysregulation of O-GlcNAcylation has been implicated in many diseases including cancer, diabetes, cardiovascular and neurodegenerative diseases[5,6]. The *Arabidopsis* genome encodes two OGT homologs: SPINDLY (SPY) and SECRET AGENT (SEC). The *spy* mutant was identified as a gibberellin (GA) response mutant with phenotypes of enhanced seed germination, early flowering, increased stem elongation, and hyposensitivity to the stress hormone abscisic acid (ABA)[7,8]. The *sec* mutants show no dramatic phenotype, but the double loss-of-function *spy sec* mutants are embryo lethal[9]. SEC and SPY were recently reported to have O-GlcNAc and O-fucosyl transferase activities, respectively, and they antagonistically regulate DELLAs, the repressors of GA signaling[10]. The lethal phenotype of *spy sec* double mutants suggests that SPY and SEC have broader functions, which remain to be investigated at the molecular level[10–12]. Our recent study identified the first large set of 971 O-GlcNAcylated peptides in 262 *Arabidopsis* proteins[13]. The functions of these O-GlcNAcylation events remain to be characterized.

One of the O-GlcNAcylated proteins is AtACINUS, an *Arabidopsis* homolog of the mammalian apoptotic chromatin condensation inducer in the nucleus (Acinus)[14]. In animals, Acinus forms the apoptosis and splicing-associated protein (ASAP) complex by recruiting RNA-binding protein S1 (RNPS1), a peripheral splicing factor, and Sin3-associated protein of 18 kDa (SAP18), a chromatin remodeler, through its conserved RNPS1–SAP18 binding (RSB) domain[14]. Another RSB-containing protein, Pinin, forms a similar protein complex named PSAP, which has distinct biological functions[14,15]. The ASAP and PSAP complexes are believed to function at the interface between histone modification, transcription, and alternative splicing (AS) in metazoans[14,16,17]. In *Arabidopsis*, AtRNPS1, also known as ARGININE/SERINE-RICH 45 (SR45), has been implicated in splicing, transcription, and RNA-dependent DNA methylation, with effects on multiple aspects of plant development as well as stress and immune responses[18–23]. AtSAP18 has been shown to associate with transcription factors involved in stress responses and embryo development[24,25]. AtACINUS, AtSAP18, and SR45 have been shown to associate with a transcription factor involved in flowering[26]. While sequence analysis predicted similar ASAP complex in plants[23], interactions among SR45, AtSAP18, and AtACINUS remain to be tested experimentally and the functions of AtACINUS and AtPININ remain to be characterized genetically.

Our finding of O-GlcNAcylation of AtACINUS suggests that the functions of AtACINUS are regulated by O-linked glycosylation[13]. We therefore performed genetic, genomic, and proteomic experiments to understand the functions of AtACINUS and its regulation by O-linked sugar modifications. Our results demonstrate key functions of AtACINUS and its distant homolog AtPININ in regulating seed germination, ABA sensitivity, and flowering, through direct involvement in AS of two key components of the ABA signaling pathway and in the transcriptional regulation of the floral repressor *FLC*. Our results further show that AtACINUS is modified by both O-GlcNAc and O-fucose, is part of the ASAP complex, and associates with splicing and transcription factors. A subset of AtACINUS-dependent AS events is altered in the *spy* and *sec* mutants, providing genetic evidence for regulation of AS by the O-linked glycosylations.

## Results

**AtACINUS and AtPININ play genetically redundant roles.** The *Arabidopsis* AtACINUS (AT4G39680) protein is 633 amino-acid long, and it shares sequence similarity to all the known motifs of the human Acinus including the N-terminal SAF-A/B, Acinus, and PIAS (SAP) motif, the RNA-recognition motif (RRM), and the C-terminal RSB motif (Fig. 1a and Supplementary Fig. 1a)[14,16,27]. *AtACINUS* is a unique gene in *Arabidopsis* with no homolog detectable using standard BLAST (Basic Local Alignment Search Tool) search of the *Arabidopsis* protein database. However, another *Arabidopsis* gene (AT1G15200, *AtPININ*) encodes a protein with an RSB domain and is considered a homolog of mammalian Pinin[14]. AtACINUS and AtPININ share 12 amino acids within the 15-amino acid region of the RSB motif (Fig. 1b), but no sequence similarity outside this motif.

To study the biological function of AtACINUS, we obtained two mutant lines that contain T-DNA insertions in the exons of *AtACINUS*, Salk_078554 and WiscDsLoxHs108_01G, which are designated *acinus-1* and *acinus-2*, respectively (Fig. 1c). These mutants showed no obvious morphological phenotypes except slightly delayed flowering (Fig. 1d, e). The weak phenotype of *acinus* is surprising considering the important function of its mammalian counterpart and the absence of any close homolog in *Arabidopsis*.

We did not expect AtACINUS and AtPININ to have redundant functions, considering their very limited sequence similarity and the fact that mammalian Acinus and Pinin have distinct functions[14]. AtPININ shares extensive sequence similarity with human Pinin surrounding the RSB domain[14] (Supplementary Fig. 1b). Phylogenetic analysis indicated that AtPININ and human Pinin belong to one phylogenetic branch that is distinct from that of AtACINUS and human Acinus (Supplementary Fig. 1c), suggesting independent evolution of ACINUS and PININ before the separation of the metazoan and plant kingdoms. However, Pinin can, through its RSB domain, interact with RNPS1 and SAP18 to form a complex (PSAP) similar to the ASAP complex. Therefore, we tested the possibility that the weak phenotype of *Arabidopsis acinus* mutants is due to functional redundancy with AtPININ.

We obtained a T-DNA insertion mutant of AtPININ (*pinin-1*, T-DNA line GABI_029C11). The *pinin-1* mutant also showed no obvious morphological phenotype (Fig. 1d). We then crossed *pinin-1* with *acinus-1* and *acinus-2* to obtain double mutants. Both *acinus-1 pinin-1* and *acinus-2 pinin-1* double mutants displayed pleiotropic phenotypes including severe dwarfism, short root, pale leaves, narrow and twisted rosette leaves with a serrated margin, severely delayed flowering, altered phyllotaxis, increased numbers of cotyledons and petals, and reduced fertility (Fig. 1d, e and Supplementary Fig. 2). The *acinus-2 pinin-1* double mutants transformed with *35S::AtACINUS-GFP* or *35S::YFP-AtPININ* displayed near wild-type (WT) morphology (Fig. 1f), confirming that the phenotypes of the double mutants are due to loss of both AtACINUS and AtPININ, and the two genes play genetically redundant roles. The AtACINUS-GFP and YFP-AtPININ proteins are localized in the nucleus outside the nucleolus (Supplementary Fig. 3).

We also noticed that the seed germination was delayed in the *acinus pinin* mutant (Fig. 2a). This, together with the pale leaf and dwarfism phenotypes, suggests an alteration in ABA response. Indeed, on 0.25 µmol/L ABA, germination of the *acinus-2 pinin-1* double mutant seeds was further delayed compared to the WT and the single mutants (Fig. 2b). Dose response experiment

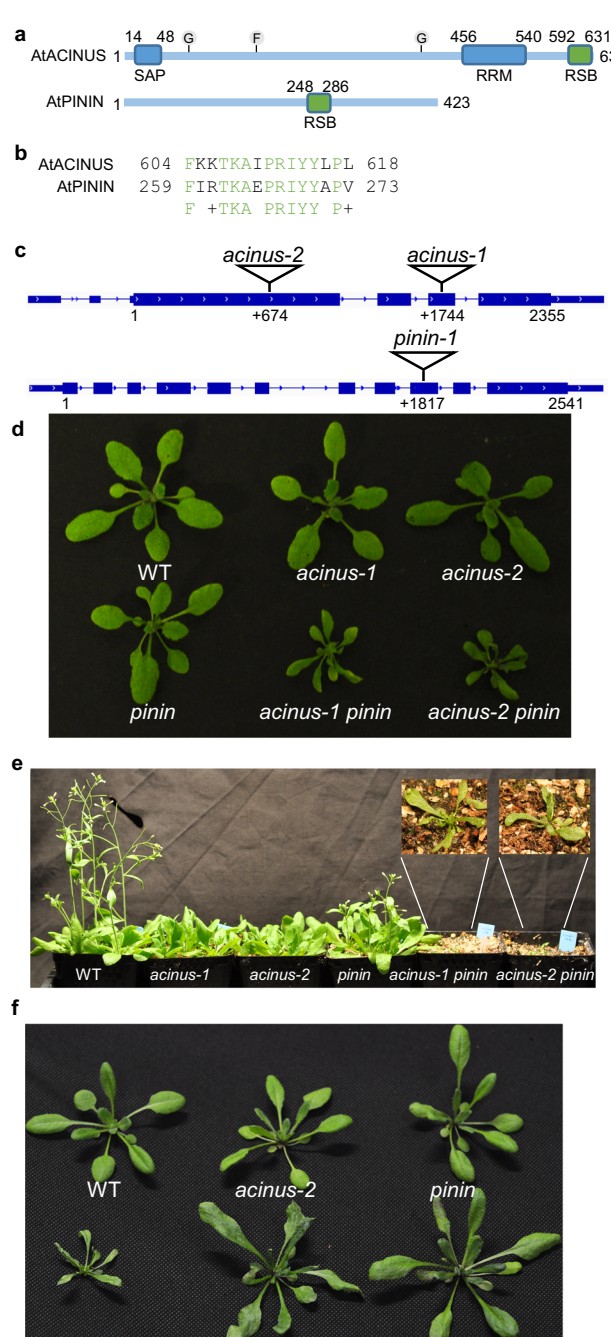

**Fig. 1 AtACINUS and AtPININ are genetically redundant. a** Diagrams of the domain structures of AtACINUS and AtPININ. SAP: SAF-A/B, Acinus, and PIAS motif. RRM: RNA-recognition motif. RSB: RNPS1–SAP18 binding domain. G and F indicates the position of O-GlcNAcylation and O-fucosylation modifications, respectively. **b** The sequence alignment of the RSB domains of AtACINUS and AtPININ. Conserved amino acids are highlighted in green. **c** Diagrams of the AtACINUS and AtPININ (translation start at position 1) with T-DNA insertion sites in *acinus-1*, *acinus-2*, and *pinin-1* mutants. **d** Plant morphologies of wild-type (WT), *acinus-1*, *acinus-2*, *pinin-1*, *acinus-1 pinin-1*, and *acinus-2 pinin-1* grown on soil for 20 days. **e** Five-week-old WT, *acinus-1*, *acinus-2*, *pinin-1*, *acinus-1 pinin-1*, and *acinus-2 pinin-1* plants grown under long day condition. Inset shows enlarged view of the *acinus*-1 *pinin-1* and *acinus-2 pinin-1* mutants. **f** Expression of either AtACINUS-GFP or YFP-AtPININ suppresses the growth defects in *acinus-2 pinin-1* double mutant (*ap*).

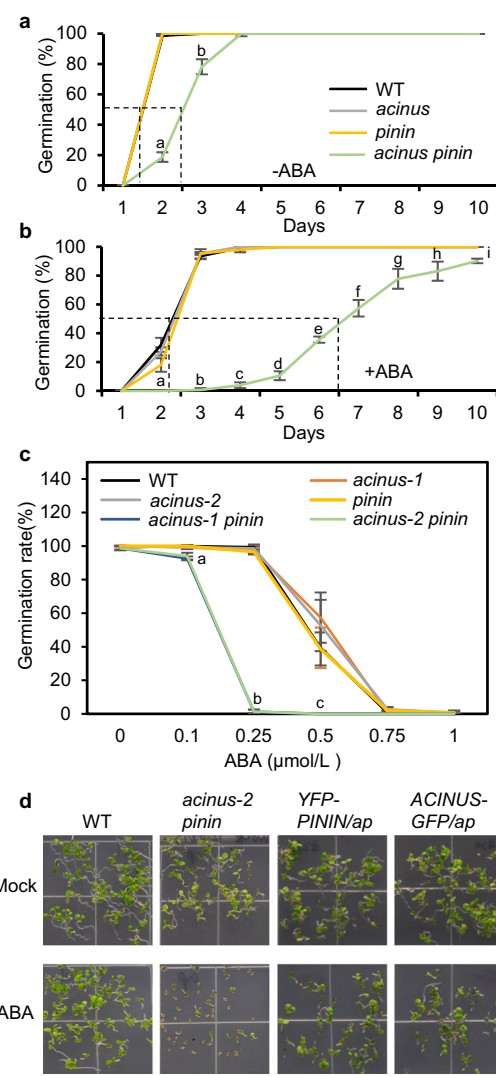

**Fig. 2 The *acinus pinin* double mutants showed ABA hypersensitive phenotypes. a, b** Germination rates of WT, *acinus-2*, *pinin-1*, and *acinus-2 pinin-1* after different days on ½ MS medium without ABA (**a**) or with 0.25 μmol/L ABA (**b**). The data points of WT, *acinus-2*, and *pinin-1* overlap. Statistically significant differences between WT and *acinus-2 pinin-1* were determined by two-tailed *t* test. The *P* values for a and b in **a** are 1.24E−5 and 2.96E−2. The *P* values for a, b, c, d, e, f, g, h, and i in **b** are 7.69E−3, 5.60E−6, 2.88E−6, 3.92E−4, 3.17E−4, 6.00E−3, 3.10E−2, 4.77E−2, and 8.57E−3. **c** Seed germination rates of the indicated genotypes on ½ MS medium supplemented with increasing concentrations of ABA after 5 days. Note that the data points of *acinus-1 pinin-1* and *acinus-2 pinin-1* overlap and those of WT, *acinus-1*, *acinus-2*, and *pinin-1* overlap. Statistically significant differences between WT and *acinus-2 pinin-1* were determined by two-tailed *t* test. The *P* values for a, b, and c are 3.51E−2, 5.14E−8, and 2.99E−2. **d** Seed germination and development of the indicated genotypes on ½ MS medium with or without 0.5 μmol/L ABA. The pictures were taken 6 days after germination. Values represent mean ± SD calculated from three biological replicates (*n* = 3) for **a–c**.

indicates that seed germination of the *acinus-1 pinin-1* and *acinus-2 pinin-1* double mutants is about threefold more sensitive to ABA than WT and the *acinus* and *pinin* single mutants (Fig. 2c). Similarly, post-germination seedling growth of *acinus-2 pinin-1* was more inhibited by ABA (Supplementary Fig. 4a). These ABA-hypersensitive phenotypes were rescued by expression of either AtACINUS-GFP or YFP-AtPININ in the *acinus-2*

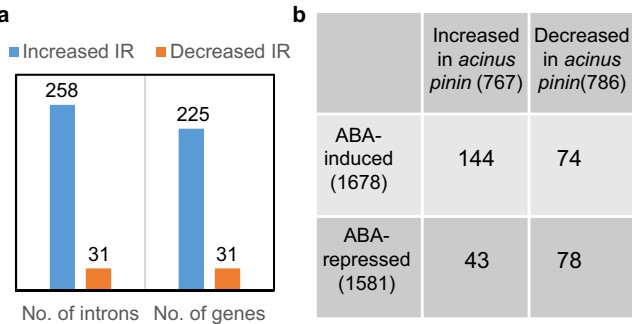

**Fig. 3 RNA-sequencing analysis of *acinus-2 pinin-1* showed differential intron retention and expression level of many genes. a** Number of introns that showed increased or decreased intron retention in *acinus-2 pinin-1* and the number of genes that contain these introns. **b** Comparison between genes differentially expressed in *acinus-2 pinin-1* and ABA-responsive genes. RNA-seq was conducted using 14-day-old light-grown seedlings for both genotypes.

*pinin-1* background (Fig. 2d and Supplementary Fig. 4b). These results indicate that the *acinus-2 pinin-1* double mutant is hypersensitive to ABA, and that AtACINUS and AtPININ are redundant negative regulators of ABA responses.

**AtACINUS and AtPININ are involved in AS of specific introns.** We conducted an RNA-seq analysis of the transcriptome of the *acinus-2 pinin-1* double mutant. WT and *acinus-2 pinin-1* seedlings were grown under constant light for 14 days, and RNA-seq was performed with three biological replicates, each yielding a minimum of 22.4 million uniquely mapped reads. The RNA-seq data confirmed the truncation of the *AtACINUS* and *AtPININ* transcripts in the double mutant (Supplementary Fig. 5). Compared to WT, the *acinus-2 pinin-1* double mutant showed significantly decreased expression levels for 786 genes and increased levels of 767 genes (fold change >2, multiple-testing corrected *p*-value <0.05), which include the flowering repressor *FLC*[28] (Supplementary Data 1).

A significantly higher proportion of reads was mapped to the intron regions in the *acinus-2 pinin-1* double mutant than in the WT (Supplementary Fig. 6a). Further analyses using the RACKJ software package revealed an increase of retention of 258 introns in 225 genes and decreased retention of 31 introns in 31 genes in the *acinus-2 pinin-1* double mutant compared to WT (Fig. 3a and Supplementary Data 2). Intron retention was the dominant form of splicing defect in the *acinus-2 pinin-1* double mutant (Fig. 3a and Supplementary Fig. 6b). About 99% of these genes contain multiple introns, and the defects tend to be retention of a specific single intron among many introns of each gene, indicating defects in AS rather than general splicing. Among the RNAs showing increased intron retention, 26 RNAs also showed decreased levels of RNA abundance, and their retained introns introduce in-frame stop codons (Supplementary Fig. 7), consistent with non-sense-mediated decay[29]. The results show that AtACINUS and AtPININ function in AS, primarily by enhancing splicing of a specific intron among many introns of each transcript.

We found a significant overlap between ABA-induced genes and the genes overexpressed in *acinus-2 pinin-1* (*p*-value by random chance <2.42E−13) (Fig. 3b). Only four of these RNAs were mis-spliced in *acinus-2 pinin-1*. One possibility is that intron retention in RNAs encoding components of ABA synthesis or signaling pathway leads to expression of ABA-responsive genes. Indeed, we found retention of the 10th intron of *ABA HYPERSENSITIVE 1* (*ABH1*) in the *acinus-2 pinin-1* double mutant (Fig. 4a).

*ABH1* encodes the large subunit of the dimeric *Arabidopsis* mRNA cap-binding complex (NUCLEAR CAP-BINDING PROTEIN SUBUNIT 1, CBP80) and functions as a negative regulator of ABA responses including inhibition of seed germination[30,31]. The retention of the 10th intron of ABH1 introduces a premature stop codon that truncates the C-terminal 522 amino acids of ABH1 (Fig. 4a). Quantification using qRT-PCR analysis in 12-day-old seedlings showed that the intron-containing *ABH1.2* transcript was about 8–10% of the total *ABH1* transcripts in the WT, about 11% in *pinin-1*, about 15% in *acinus-2*, but more than 50% in *acinus-2 pinin-1* (Fig. 4b, c). Expression of either YFP-AtPININ or AtACINUS-GFP in the *acinus-2 pinin-1* background rescued the *ABH1* intron retention phenotype (Fig. 4b, c). Consistent with compromised *ABH1* activity, the gene expression changes in *acinus-2 pinin-1* show a strong correlation to those in *abh1*, with Spearman's correlation = 0.74 as calculated by AtCAST3.1 (Supplementary Fig. 8)[32,33].

Intron retention in *HAB1* has been reported to cause ABA hypersensitive phenotypes[34,35]. *HAB1* did not display any apparent splicing defects in our RNA-seq and RT-PCR analysis of the 12-day-old seedling. However, after ABA treatment, *HAB1* intron retention is significantly increased in *acinus pinin* compared to the WT. While the expression level of *HAB1* transcripts was increased similarly in WT and *acinus pinin*, the WT seedlings maintained relatively similar ratios between different splice forms of *HAB1* before and after ABA treatment, whereas the *acinus pinin* mutant accumulated a much increased level of the intron-containing *HAB1.2* and a reduced level of fully spliced *HAB1.3* after ABA treatment (Fig. 4d, e). *HAB1.2* encodes a dominant negative form of HAB1 protein that activates ABA signaling[34,35]. Therefore, the accumulation of HAB1.2 should contribute to the ABA hypersensitivity of the *acinus pinin* mutant.

To test whether AtACINUS is directly involved in AS of *ABH1* and *HAB1*, we carried out an RNA immunoprecipitation (RNA-IP) experiment using an *AtACINUS-GFP/acinus-2* transgenic line, with *35S::GFP* transgenic plants as the negative control. Immunoprecipitation using an anti-GFP antibody pulled down significantly more *ABH1* and *HAB1* RNAs in *AtACINUS-GFP/acinus-2* than in the *35S::GFP* control (Fig. 4f, g), indicating that AtACINUS interacts with *ABH1* and *HAB1* RNAs in vivo and is involved in their splicing.

**AtACINUS regulates flowering through repression of *FLC*.** Consistent with the late-flowering phenotype of *acinus pinin* (Figs. 1e, 5a), our RNA-seq data showed an increased expression level of the floral repressor *FLC*, without obvious alteration of the splicing pattern (Supplementary Fig. 9a). The RT-qPCR analysis confirmed the increased levels of *FLC* RNA that correspond to the severity of the late-flowering phenotypes in the single and double mutants (Fig. 5b). As *FLC* expression is also controlled by its anti-sense RNA, which undergoes AS[36,37], we analyzed the anti-sense *FLC* RNAs using RT-qPCR. The results showed a dramatic increase of the class I anti-sense RNA and a slight increase of the class II anti-sense RNA of *FLC*, but no obvious change of the splicing efficiency of the *FLC* anti-sense RNAs (Supplementary Fig. 9b–d). AtACINUS was recently reported to associate with VAL1 and VAL2, which bind to the *FLC* promoter to repress transcription[26]. We thus performed chromatin immunoprecipitation (ChIP) assays to test whether AtACINUS is associated with the *FLC* locus, and our results show that AtACINUS interacts with the DNA of the promoter and first intron regions but not the 3′ region of *FLC* in vivo (Fig. 5c). Together our results provide evidence for a role of AtACINUS in regulating the transcription of *FLC*.

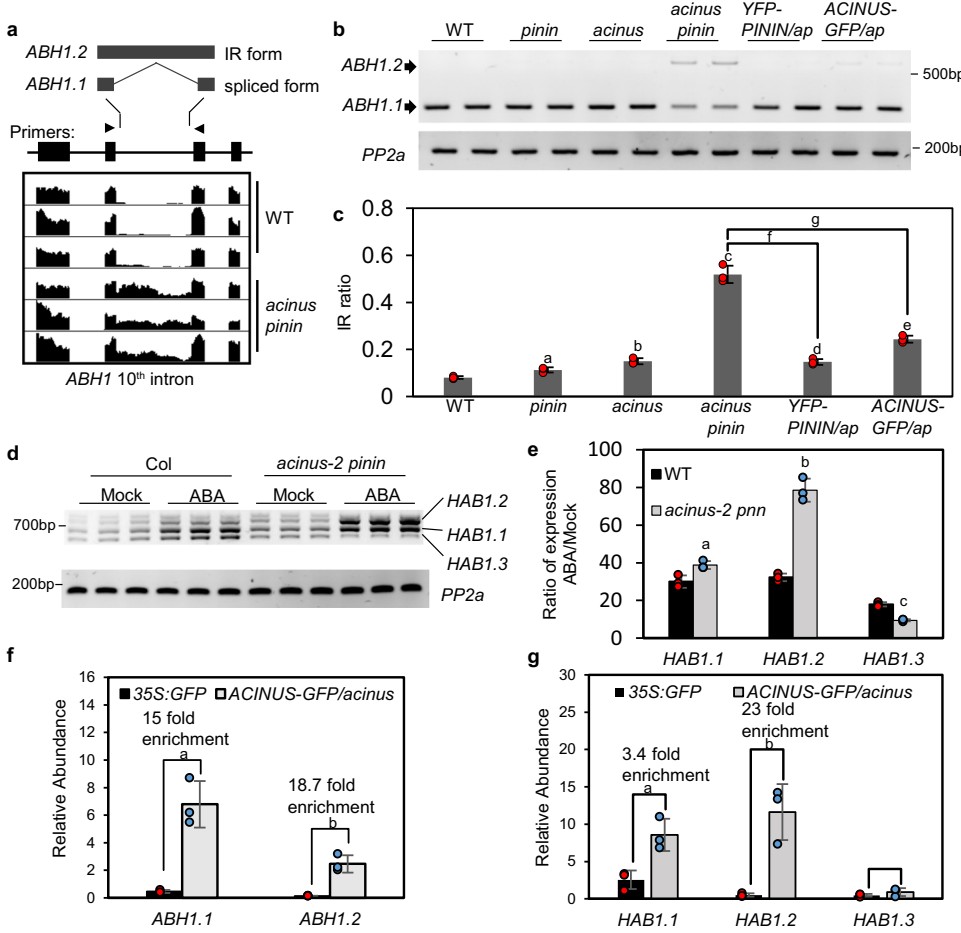

**Fig. 4 *ABH1* and *HAB1* showed increased intron retention in *acinus-2 pinin-1* and *ABH1* and *HAB1* mRNAs are associated with AtACINUS. a** Integrative genomic viewer (IGV) display of increased intron retention (IR) of the *ABH1* 10th intron in *acinus-2 pinin-1* compared to WT. **b** RT-PCR of *ABH1* in 12-day-old seedlings of the indicated genotypes using primers at positions indicated by arrowheads in **a**. **c** Intron retention ratio of *ABH1* 10th intron as determined by RT-qPCR in 12-day-old seedlings of the indicated genotypes. The intron-containing form *ABH1.2* was highly accumulated while the spliced form *ABH1.1* was reduced in *acinus-2 pinin-1* compared to WT, the single mutants, or the double mutant complemented by *YFP-AtPININ* or *AtACINUS-GFP*. Statistically significant differences to WT or between indicated genotypes were determined by two-tailed *t* test. The *P* values for a, b, c, d, e, f, and g are 1.97E−2, 5.02E−3, 1.94E−3, 4.90E−4, 9.01E−4, 1.39E−3, and 2.32E−3. **d** RT-PCR of *HAB1* in 12-day-old WT and *acinus-2 pinin-1* seedlings treated with ABA (100 μmol/L for 3 h). **e** RT-qPCR quantification of the fold changes of expression levels of each splice forms of *HAB1* after ABA treatment of 12-day-old WT and *acinus-2 pinin-1* seedlings. The *P* values for a, b, and c are 2.29E−2, 3.08E−3, and 1.60E−3. **f, g** Quantification of *ABH1* and *HAB1* mRNAs by qPCR after RNA-IP using α-GFP antibody in 7-day-old *AtACINUS-GFP/acinus-2* seedlings, compared to *35S::GFP* as a negative control. Statistical significance was determined by two-tailed *t* test. The *P* values for a and b in **f** are 2.24E−2 and 2.31E−2. The *P* values for a and b in **g** are 2.18E−2 and 3.54E−2. Values represent mean ± SD calculated from three biological replicates (*n* = 3) for **c**, **e**, **f**.

**AtACINUS-dependent AS events are altered in *spy* and *sec*.** To study how O-linked sugar modification affects the function of AtACINUS, we tested if the AtACINUS-dependent AS events are altered in the *spy* and *sec* mutants. Of the ten AtACINUS-dependent intron splicing events we have tested, four showed alterations in the *spy* mutant and one showed alteration in the *sec* mutant (Fig. 6).

In the 7-day-old light-grown plants, splicing of the 12th intron and the 15th intron of *TRNA METHYLTRANSFERASE 4D* (*TRM4D, At4g26600*) was enhanced in the *acinus-2 pinin* double mutant compared to that in the WT. In the loss-of-function mutants *spy-4* and *spy-t1* (SALK_090580), the splicing efficiency of these two introns were also enhanced. In contrast, the loss-of-function mutants *sec-2* and *sec-5* showed an increased retention of the 12th intron (Fig. 6). These results suggest that SPY and SEC have opposite effects on AtACINUS function in *TRM4D* splicing. The *spy-t1* and *spy-4* mutants accumulated more *HAB1.3* and less *HAB1.2* than WT, while *acinus-2 pinin* accumulated more

*HAB1.2* than the WT (Fig. 6), consistent with their opposite seed germination phenotypes. In addition, the splicing efficiency of the 14th intron of *EMBRYO DEFECTIVE 2247* (*Emb2247, AT5G16715*) was reduced in the *acinus-2 pinin* double mutant, but was increased in the *spy-t1* and *spy-4* mutants compared to WT (Fig. 6). These results support that the O-linked sugar modifications of AtACINUS modulate its functions in AS of specific RNAs.

**AtACINUS associates with transcriptional and splicing factors.** To understand the molecular mechanisms of AtACINUS function, we conducted two immunoprecipitations followed by mass spectrometry (IP-MS) experiments. In the first experiment, immunoprecipitation was performed in three biological replicates using the AtACINUS-GFP/*acinus-2* plants and the anti-GFP nanobody. Transgenic plants expressing a Tandem-Affinity-Purification-GFP (TAP-GFP) protein were used as control[38].

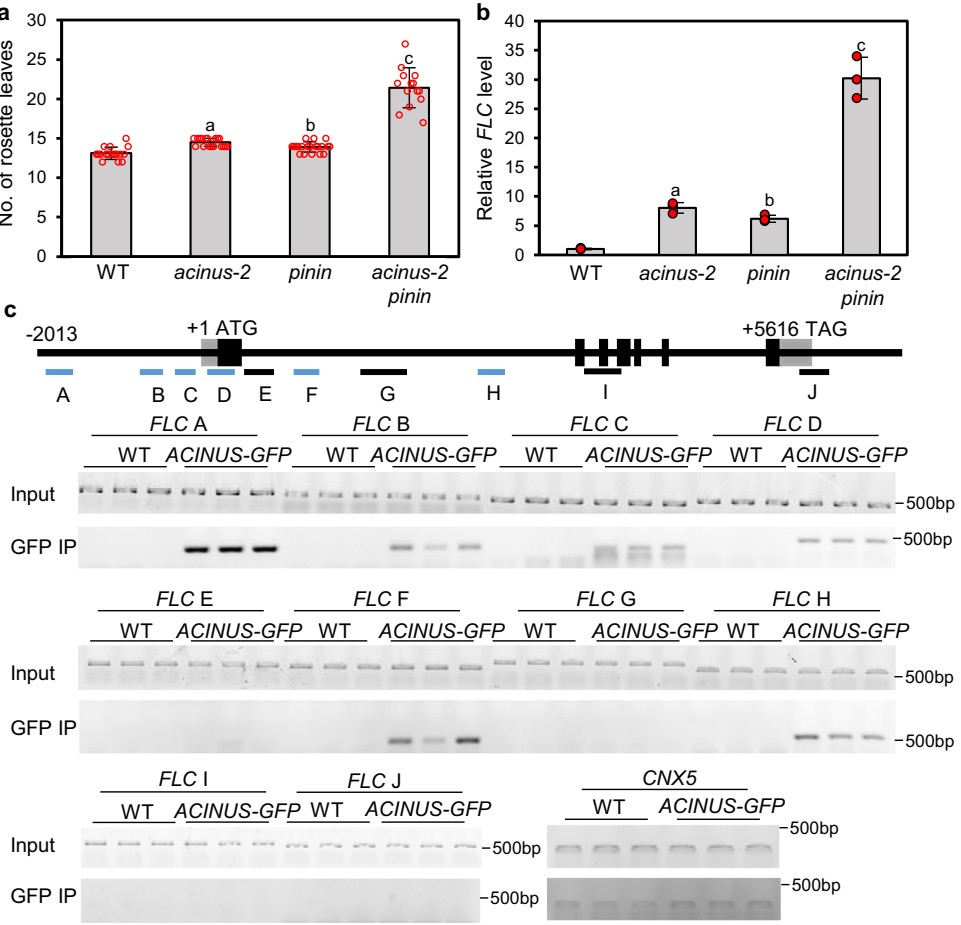

**Fig. 5 The *acinus-2 pinin-1* double mutant is late flowering with increased *FLC* expression. a** Rosette leaf numbers of WT, *acinus-2*, *pinin-1*, and *acinus-2 pinin-1* at bolting stage grown in long day condition. Error bars indicate SD calculated from $n > 12$. Values represent mean ± SD calculated from at least 12 plants ($n > 12$). Statistically significant differences to WT were determined by two-tailed $t$ test. The $P$ values for a, b, and c are 1.35E−6, 2.48E−3, and 5.15E−9. **b** *FLC* expression level relative to *UBQ (At5g15400)* in WT, *acinus-2*, *pinin-1*, and *acinus-2 pinin-1*, determined by RT-qPCR in 12-day-old seedlings. Values represent mean ± SD calculated from three biological replicates ($n = 3$). Statistically significant differences to WT were determined by two-tailed $t$ test. The $P$ values for a, b, and c are 5.15E−3, 3.32E−3, and 4.91E−3. **c** Analysis of AtACINUS-GFP association with the *FLC* locus by ChIP-PCR in 12-day-old *AtACINUS-GFP/acinus-2* seedlings. WT serves as the negative control. Bars below the gene structure diagram represent regions analyzed by PCR (blue bars indicate regions enriched after immunoprecipitation). GFP IP shows PCR products using immunoprecipitated DNA. *CO-FACTOR FOR NITRATE, REDUCTASE, AND XANTHINE DEHYDROGENASE 5* (*CNX5*) serves as an internal control to show non-specific background DNA after immunoprecipitation. PCR reactions were set to 28 cycles.

The proteins co-immunoprecipitated with AtACINUS-GFP were identified based on enrichment (FDR = 0.01, S0 = 2) relative to the TAP-GFP control, quantified by label-free mass spectrometry analysis. In the second experiment, AtACINUS-associated proteins were identified by [15]N stable-isotope-labeling in *Arabidopsis* (SILIA) quantitative mass spectrometry. WT and *acinus-2* mutant seedlings were metabolically labeled with [14]N and [15]N, and immunoprecipitation was performed using the anti-AtACINUS antibody, followed by mass spectrometry analysis. The isotope labels were switched in the two biological replicates. AtACINUS-associated proteins were identified based on enrichment in the WT compared to the *acinus* mutant control. These IP-MS experiments consistently identified 46 AtACINUS-associated proteins (Fig. 7a, Supplementary Fig. 10a, and Supplementary Data 3). These included SR45 and AtSAP18, supporting the existence of an evolutionarily conserved ASAP complex in *Arabidopsis*. The AtACINUS interactome also included a large number of proteins homologous to known components of the spliceosome, including five Sm proteins, one protein of the U2 complex, four proteins in the U5 complex, 17 proteins of the nineteen complex (NTC) and NTC-related complex (NTR)[39–41].

In addition, AtACINUS associated with six proteins of the exon junction complex (EJC) core and the EJC-associated TRanscription-EXport (TREX) complex, three proteins of the small nucleolar ribonucleoprotein (snoRNP) complexes, and four other splicing-related proteins (Fig. 7a and Supplementary Data 3)[41–45]. AtACINUS interactome also included a component of the RNA Polymerase II Associated Factor 1 complex (PAF1C) (Fig. 7a and Supplementary Data 3). The interactome data suggest that, similar to mammalian Acinus, AtACINUS plays dual roles in AS and transcriptional regulation.

The AtACINUS interactome includes five proteins that are genetically involved in regulating *FLC* and flowering (Fig. 7a and Supplementary Data 3). These are BRR2 and PRP8 of the U5 complex, ELF8 of the PAF1C, and SR45 and AtSAP18 of the ASAP complex[19,37,46,47]. These results suggest that AtACINUS may regulate *FLC* expression through a complex protein network involving multiple regulatory pathways.

We have previously identified O-GlcNAc modification on Thr79 on AtACINUS[13] (Fig. 7b) after LWAC enrichment. Mass spectrometry analysis following affinity purification of AtACINUS identified additional O-GlcNAc modification on the peptide

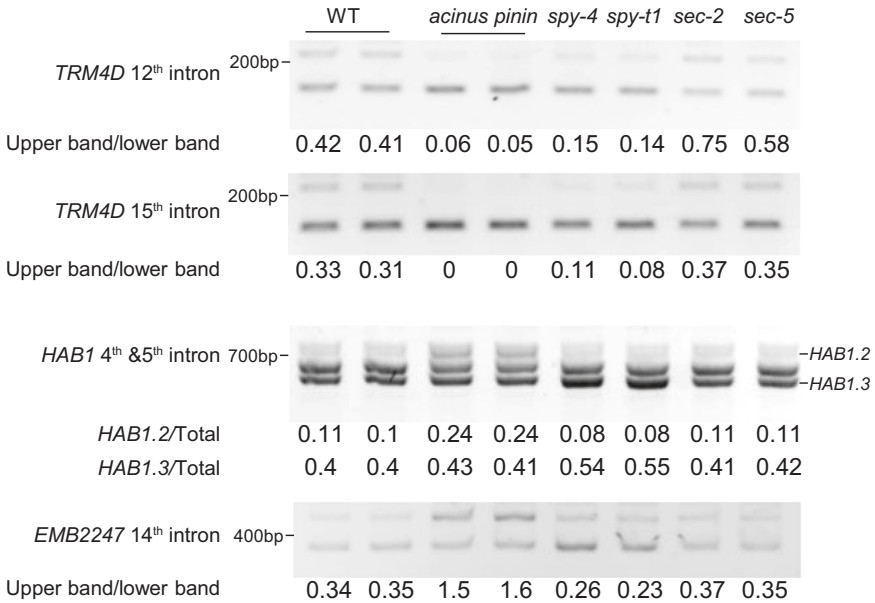

**Fig. 6 A subset of AtACINUS-dependent intron splicing events are affected in the *spy* and *sec* mutants.** RT-PCR of *HAB1*, *EMB2247*, and *TRM4D* in 7-day-old WT, *acinus-2 pinin*, *spy-t1*, *spy-4*, *sec-2*, and *sec-5* seedlings with primers flanking the targeted introns.

containing amino acids 407–423 (Fig. 7c and Supplementary Fig. 10b), as well as O-fucosylation on the peptide containing amino acids 169–197 (Fig. 7d). These results confirm that AtACINUS is a target of both O-GlcNAc and O-fucose modifications.

Using targeted mass spectrometry analysis, we confirmed that the *acinus-2 pinin* double mutant expressed only the AtACINUS's N-terminal peptides (at about 20% WT level), but no detectable peptides of the C-terminal region (after T-DNA insertion) (Supplementary Fig 11 and Supplementary Data 5). Both N- and C-terminal peptides of AtPININ were undetectable in the *acinus-2 pinin* mutant (Supplementary Fig 12 and Supplementary Data 5). Meanwhile, SR45 and AtSAP18 protein levels were dramatically reduced to 3.9% and 2.7% of WT levels, respectively (Supplementary Figs. 13 and 14 and Supplementary Data 5). Together, these results indicate that the stability of the other members of the ASAP and PSAP complexes is dependent on AtACINUS and AtPININ.

## Discussion

Our recent identification of O-GlcNAcylated proteins in *Arabidopsis* enabled functional study of this important signaling mechanism in plants[13]. Here our systematic analysis of one of these O-GlcNAcylated proteins, AtACINUS, demonstrates its functions as a target of O-GlcNAc and O-fucose signaling and a component of the evolutionarily conserved ASAP complex that regulates transcription and RNA AS thereby modulating stress responses and developmental transitions. Our comprehensive genetic, transcriptomic, and proteomic analyses provide a large body of strong evidence illustrating a molecular pathway in which nutrient sensing O-GlcNAcylation and O-fucosylation modulate specific functions of the evolutionarily conserved RSB-domain protein AtACINUS to modulate stress hormone sensitivity, seed germination, and flowering in plants (Fig. 7e).

Studies in animals have identified Acinus and Pinin as essential cellular components that bridge chromatin remodeling, transcription, and splicing through the formation of analogous ASAP and PSAP complexes[14,16,17,48–51]. Sequence alignment and phylogenetic analysis show that the *Arabidopsis* orthologs,

AtACINUS and AtPININ, share higher levels of sequence similarity to their animal counterparts than to each other and appear to have evolved independently since the separation of the plant and metazoan kingdoms[14]. Considering their evolutionary distance and limited sequence similarity (12 amino acid residues in the RSB motif), it was surprising that the functions of AtACINUS and AtPININ are genetically redundant. This represents likely the least sequence similarity between two redundant genes and raises cautions for prediction of genetic redundancy based on the level of sequence similarity.

The developmental functions in seed germination and flowering seem to involve AtACINUS's distinct activities in splicing and transcription of key components of the regulatory pathways. Specifically, AS events in *ABH1* and *HAB1* are likely the major mechanisms by which AtACINUS modulates ABA signaling dynamics to control seed germination and stress responses. ABH1 is an mRNA cap-binding protein that modulates early ABA signaling[30,31]. The loss-of-function *abh1* mutant with a T-DNA insertion in the 8th intron is ABA hypersensitive with enhanced early ABA signaling[30]. Similarly, the retention of the 10th intron of *ABH1* in *acinus pinin* mutant is expected to truncate its C-terminal half and cause loss of *ABH1* function and thus increase of ABA sensitivity. Supporting the functional role of the ASAP/PSAP-ABH1 pathway, we observed a significant correlation between the transcriptomic changes in *abh1* and the *acinus pinin* double mutant (Supplementary Fig. 8)[32,33]. A recent proteomic study showed that the ABH1 protein level was decreased in the *sr45* mutant[23], whereas a reduction of *ABH1* RNA level to ~30% caused obvious phenotypes in potato[52].

AtACINUS-mediated AS of *HAB1* switches a positive feedback loop to a negative feedback loop in the ABA signaling pathway. *HAB1* encodes a phosphatase that dephosphorylates the SNF1-related protein kinases (SnRK2s) to inhibit ABA responses, and the ligand-bound ABA receptor inhibits HAB1 to activate ABA responses[53,54]. The intron-containing *HAB1.2* encodes a dominant negative form of HAB1 protein that lacks the phosphatase activity but still competitively interacts with SnRK2, thus activating, instead of inhibiting, ABA signaling[34,35]. As ABA signaling feedback increases the *HAB1* transcript level, the AtACINUS-mediated AS switches a positive feedback loop that

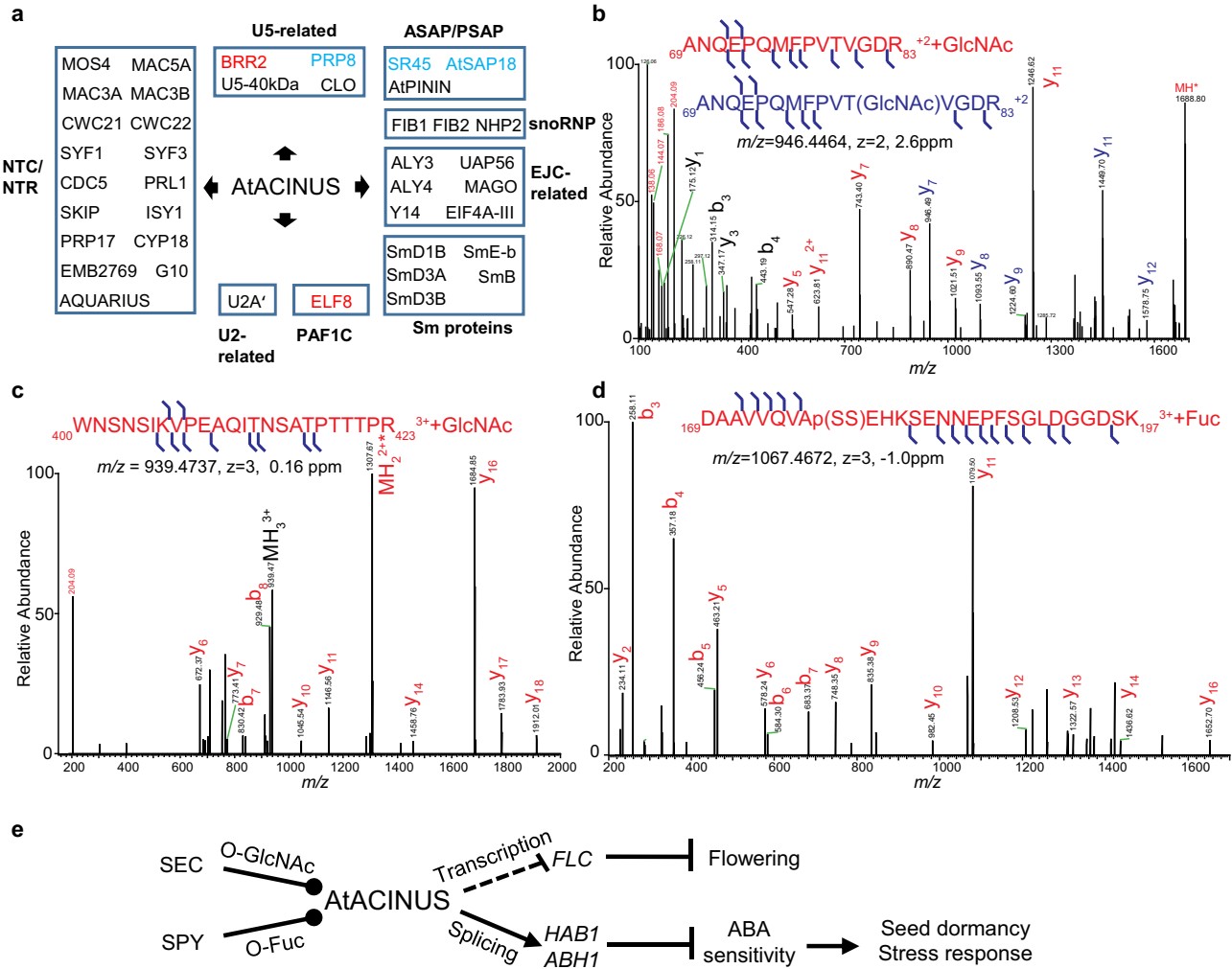

**Fig. 7 AtACINUS is O-GlcNAc and O-fucose modified and associates with spliceosomal complexes, transcriptional regulators, and chromatin remodeling proteins. a** Diagram shows functional groups of AtACINUS-associated proteins. Proteins are grouped in boxes based on their association with known complexes or functions. Positive regulators of *FLC* are highlighted in red and negative regulators in blue. Seven-day-old seedlings were used for the label-free IP-MS experiments and 14-day-old seedlings were used for the [15]N stable-isotope-labeling in *Arabidopsis* (SILIA) quantitative MS experiments. **b**, **c** Higher energy collisional dissociation (HCD) mass spectra show O-GlcNAcylation on Thr79 and a sequence spanning amino acid 400–423 of AtACINUS. The sequence ion series that retain this modification (shifted by 203 Da) are labeled in blue (**b**). The sequence ion series that have lost the modification are labeled in red. HexNAc oxonium ion (*m/z* 204) and its fragments masses are labeled in red. **d** HCD spectrum shows O-fucosylation on a sequence spanning amino acid 169–197 of AtACINUS with neutral loss. **e** Proposed model of a molecular pathway in which nutrient sensing O-GlcNAcylation and O-fucosylation modulate the evolutionarily conserved RSB-domain protein AtACINUS, which controls transcription and alternative RNA splicing of specific target genes to modulate stress hormone sensitivity and developmental transitions such as seed germination and flowering in plants.

reinforces ABA signaling to a negative feedback loop that dampens ABA signaling. Such a switch is presumably important for the different ABA signaling dynamics required for the onset of and recovery from stress responses or dormancy.

The relative contributions of intron retention of *ABH1* and *HAB1* to ABA sensitivity will need to be quantified by genetic manipulation of each splicing event. Additional mechanisms may contribute to the ABA-hypersensitivity phenotypes of *acinus pinin*. For example, the level of SR45 is significantly decreased in *acinus pinin*, while loss of SR45 has been reported to cause accumulation of SnRK1 which is a positive regulator of stress and ABA responses[55].

The late-flowering phenotype of the *acinus pinin* mutant correlated with increased *FLC* expression. A role of AtACINUS in repressing *FLC* has been suggested based on its association with the VAL1 transcription factor, which binds to the *FLC* promoter[26]. Our results provide genetic evidence for the function of

AtACINUS in repressing *FLC* expression. Further, our ChIP-PCR analysis shows that AtACINUS associates with genomic DNA of the promoter region and the first intron of *FLC*, confirming a direct role in transcriptional regulation of *FLC*. These results provide critical evidence for the hypothesis that the AtACINUS represses *FLC* by AtSAP18-mediated recruitment of the Sin3 histone deacetylase complex (HDAC)[26]. It is worth noting that overexpression of AtSAP18 in the *sr45* mutant increased *FLC* expression and further delayed flowering[23]. It is possible that the transcriptional repression function of AtSAP18 requires the ASAP/PSAP complex. It is also worth noting that the AtACINUS interactome includes several proteins known to be involved in regulating *FLC* expression and flowering. Among these, BRR2 and PRP8 are components of the U5 complex and mediate splicing of the sense and anti-sense transcripts of *FLC* to inhibit and promote flowering, respectively[37,46]. ELF8 is a component of the PAF1 complex and promotes histone methylation of *FLC* chromatin[47].

The identification of additional *FLC*-regulators as AtACINUS-associated proteins suggests that AtACINUS may regulate *FLC* expression through complex protein networks. Genetic evidence supports that ELF8/PAF1C and SR45 also have dual functions in regulating *FLC* expression and ABA responses[18,19,22,56], suggesting that the functions of AtACINUS in seed germination and flowering may involve overlapping protein networks.

Structural studies in metazoan systems showed that the RSB domains of Acinus and Pinin directly interact with RNPS1 and SAP18, forming a ternary ASAP and PSAP complexes that have both RNA- and protein-binding properties as well as abilities to interact with both RNA splicing machinery and histone modifiers[14]. ASAP and PSAP function as EJC peripheral protein complexes to modulate RNA processing[15,57]. Our quantitative proteomic analyses of the AtACINUS interactome provide strong evidence for interaction with SR45 (ortholog of RNPS1) and AtSAP18, as well as components of EJC and additional splicing factors. However, some proteins, such as SPY and SEC, may interact transiently and were not detected by IP-MS. While our proteomic data do not distinguish the proteins that directly interact with AtACINUS from those that associate indirectly as a subunit of the interacting protein complexes, the greatly reduced levels of SR45 and AtSAP18 proteins in *acinus pinin* are consistent with the direct interactions predicted based on the conserved RSB domain. Similarly, the *sr45* mutation leads to a near absence of AtSAP18 and a mild decrease of the AtACINUS protein level in the inflorescence tissues[23]. Together these observations support the notion that AtACINUS and AtPININ mediate formation of similar ASAP and PSAP complexes and stabilize SR45 and AtSAP18 in plants.

Studies in human cells have shown that Acinus and Pinin mediate splicing of distinct RNAs and that Acinus cannot rescue the splicing defects caused by knockdown of Pinin[15]. In contrast, AtACINUS and AtPININ appear to have largely redundant and interchangeable functions. It is possible that both AtACINUS and AtPININ, through their RSB domain, recruit SR45 and AtSAP18, which determine target specificities. However, AtACINUS and AtPININ may have subtle differences in their functions. Like human Acinus, AtACINUS contains two additional conserved domains that are absent in AtPININ. Further, the regions of AtACINUS and AtPININ, as well as human Acinus and Pinin, outside the RSB domain contain mostly divergent intrinsically disordered sequences[58] (Supplementary Fig. 15). These distinct sequences may provide specificity in interactions with target transcripts and partner proteins or in regulation by PTMs[58]. Indeed, O-GlcNAcylated residues (Thr79 and amino acids 407–423) and the O-fucosylated site (amino acids 169–197) were in the intrinsically disordered regions of AtACINUS, whereas no O-GlcNAc or O-fucose modification was detected in AtPININ, though this could be due to partial sequence coverage of our mass spectrometry analysis. Deep RNA-seq analysis with higher sequence coverage of the single and double mutants of *acinus* and *pinin* will be required to fully understand their functional overlap and specificities.

How SEC/O-GlcNAc and SPY/O-fucose modulate development and physiology of plants is not fully understood at the molecular level. The mechanism of regulating GA signaling involves antagonistic effects of O-fucosylation and O-GlcNAcylation of the DELLA proteins[10]. Similarly, we observed the opposite effects of *spy* and *sec* on the splicing of the 12th intron of *TRM4D*, suggesting distinct effects of O-GlcNAcylation and O-fucosylation on AtACINUS functions. Consistent with their different phenotype severities, more AS events were affected in *spy* than *sec*. The *spy* mutant showed increased splicing for four of the ten introns analyzed; two of these introns (in *TRM4D*) were more spliced and the other two (*HAB1* and *EMB2247*) were less

spliced in the *acinus pinin* mutant than in WT, suggesting that the SPY-mediated O-fucosylation may have different effects on AtACINUS activities on different transcripts. The two O-GlcNAc-modified residues (Thr79 and amino acids 407–423) and the O-fucose modified residue (amino acids 169–197) are in different regions of the intrinsically disordered sequence[58] (Supplementary Fig. 15), suggesting that PTMs in the disordered sequences play roles in substrate-specific splicing activities.

The high percentage of AtACINUS-dependent AS events affected in *spy* and *sec* supports an important function of AtACINUS in mediating the regulation of AS by O-glycosylation. On the other hand, AtACINUS-independent mechanisms may also contribute to the regulation, as the O-GlcNAcylated *Arabidopsis* proteins include additional RNA-binding and splicing factors[13], such as SUS2 which is in the AtACINUS interactome. Deep transcriptomic analysis of *spy*, *sec*, and conditional double *spy sec* mutants will be required to better understand how O-GlcNAc and O-fucose modulate RNA processing and AtACINUS function. Genetic analyses have suggested that SPY acts upstream of the ABA insensitive 5 (ABI5) transcription factor in regulating seed germination[8]. The molecular link between SPY/O-fucose and ABA signaling has remained unknown. Our results support a hypothesis that O-fucose modification modulates AtACINUS activity in splicing a subset of transcripts including *HAB1* to modulate ABA sensitivity. The biological function of this SPY-AtACINUS pathway remains to be further evaluated by genetic analyses including mutagenesis of the O-fucosylation sites of AtACINUS. It is likely that parallel pathways also contribute to the regulation of ABA sensitivity and seed germination by O-fucosylation and O-GlcNAcylation. For example, increased GA signaling was thought to contribute to ABA hyposensitivity in the *spy* mutant[59]. Further, the ABA response element binding factor 3 (ABF3) is also modified by O-GlcNAc[13]. The function of O-glycosylation in stress responses seems to be conserved, as large numbers of molecular connections between O-GlcNAc and stress response pathways have been reported in metazoans[5].

How O-linked glycosylation of AtACINUS affects its transcriptional activity at the *FLC* locus remains to be investigated. Both *spy* and *sec* mutants flower early, opposite to *acinus pinin*. While *spy* shows a strong early flowering phenotype, the *FLC* expression level was unaffected in *spy* under our experimental conditions (Supplementary Fig. 16), suggesting that SPY regulates flowering independent of *FLC*. The *FLC* level was decreased in *sec*[60], supporting the possibility that O-GlcNAcylation affects AtACINUS transcription activity. However, the effect of *sec* on *FLC* expression could also be mediated by other O-GlcNAc-modified flowering regulators[13,60].

Our study reveals important functions of AtACINUS in developmental transitions and a previously unknown function of O-linked glycosylation in regulating RNA AS. While we were getting our revised manuscript ready for submission, evidence was reported for a similar function of O-GlcNAc in intron splicing in metazoan and for broad presence of stress-dependent intron retention in plants. Interestingly, inhibition of OGT was found to increase splicing of detained introns in human cells[61]. Detained introns are a novel class of post-transcriptionally spliced (pts) introns, which are one or few introns retained in transcripts where other introns are fully spliced[62]. Transcripts containing pts introns are retained on chromatin and are considered a reservoir of nuclear RNA poised to be spliced and released when a rapid increase of protein level is needed, such as in neuronal activities[62,63]. A recent study uncovered a large number of pts introns in *Arabidopsis*. A significant portion of these pts introns show enhanced intron retention under stress conditions. Several splicing factors involved in pts intron splicing, MAC3A, MAC3B, and SKIP[64], are parts of the AtACINUS interactome. Among the

introns retained in the *acinus pinin* mutant, 114 are pts introns, which is about 1.7-fold the random probability (*p* value <3.0E−9). These pts introns include the intron retained in *ABH1* but not that in *HAB1*, consistent with translation of the dominant negative form of HAB1.2 (refs. [34,35]). Together with these recent developments, our study raises the possibility that AtACINUS plays important roles in the splicing of pts introns, acting downstream of the metabolic signals transduced by SPY/O-fucose and SEC/O-GlcNAc. Our study supports an evolutionarily conserved function of O-glycosylation in regulating RNA splicing, thereby linking metabolic signaling with switches of cellular status between normal and stress conditions as well as during developmental transitions.

## Methods

**Plant materials.** All the *Arabidopsis thaliana* plants used in this study were in the Col-0 ecotype background. The plants were grown in greenhouses with a 16-h light/8-h dark cycle at 22–24 °C for general growth and seed harvesting. For seedlings grown on the medium in Petri dishes, the sterilized seeds were grown on ½ Murashige and Skoog (MS) medium and supplemented with 0.7% (w/v) phytoagar. Plates were placed in a growth chamber under the constant light condition at 21–22 °C. T-DNA insertional mutants *atacinus-1* (Salk_078554, insertion position +1744 relative to the genomic translational start site of At4G39680), *atacinus-2* (WiscDsLoxHs108_01G, insertion position +674), *atpinin-1* (GABI_029C11/CS402723, insertion position +1817 of At1G15200), *spy-t1* (Salk_090580), and *sec-5* (Salk_034290) were obtained from *Arabidopsis* Biological Resource Center. The *spy-4* and *sec-2* seeds that have been backcrossed to Columbia for six generations were provided by Neil Olszewski lab.

**Germination assay.** Seeds were surface sterilized with 70% (v/v) ethanol and 0.1% (v/v) Triton X-100 sterilization solution for 5 min. The sterilization solution was then removed and seeds were resuspended in 100% ethanol and dried on a filter paper. The sterilized seeds were then plated on ½ MS medium supplemented with mock or ABA. The seeds were placed in 4 °C cold room for 3 days for stratification before moving into a growth chamber to germinate. Germination was defined as obvious radicle emergence from the seed coat.

**Gene cloning and plant transformation.** The *AtACINUS* cDNA was initially cloned into the vector pENTR-D/TOPO and subsequently into the binary vector pGWB5 to generate the *35S::AtACINUS-GFP* plasmid. The *35S::AtACINUS-GFP* binary plasmid was transformed into *acinus-2* plants by floral dipping with *Agrobacterium tumefaciens* strain GV3101. A homozygous *35S::AtACINUS-GFP/acinus-2* plant was selected for similar protein expression level to the endogenous AtACINUS protein of WT plants using a native α-AtACINUS antibody, and crossed with *acinus-2 pinin-1* to obtain *35S::AtACINUS-GFP/acinus-2 pinin-1* transgenic lines. Similarly, *35S::AtACINUS-YFP-TurboID* plasmid was generated by LR reaction of gateway-compatible *35S::YFP-TbID*[65] with *pENTR-AtACINUS* and transformed to *acinus-2 pinin-1* to obtain transgenic lines.

The *AtPININ* cDNA was acquired from the *Arabidopsis* stock center and subsequently cloned into the binary vector pEarleyGate104 to generate the *35S::YFP-AtPININ* vector. The *35S::YFP-AtPININ* binary plasmid was transformed into *acinus-2 pinin-1/+* plants by floral dipping with *A. tumefaciens* strain GV3101. Transgenic plants were genotyped for *pinin-1* allele to obtain *35S::YFP-AtPININ/acinus-2 pinin-1* transgenic lines.

**Bioinformatics analysis.** Dendrogram of AtACINUS and AtPININ homologs in different species was constructed using the "simple phylogeny" web tool of EMBL-EBI website with UPMGA method using default settings (https://www.ebi.ac.uk/Tools/phylogeny/simple_phylogeny/). The protein alignment was generated using MUSCLE from EMBL-EBI with the default setting (https://www.ebi.ac.uk/Tools/msa/muscle/)[66,67]. Pairwise protein sequence alignment was performed with Blastp from the NCBI blastp suite with *E*-value set to 0.01. (https://blast.ncbi.nlm.nih.gov/Blast.cgi?PAGE=Proteins).

Protein disorderliness was predicted based on amino acid sequences using PrDOS (http://prdos.hgc.jp/cgi-bin/top.cgi) with the default setting[68].

Gene expression correlation was analyzed with AtCAST3.1 using default settings (http://atpbsmd.yokohama-cu.ac.jp/cgi/atcast/home.cgi)[33].

**RNA sequencing and data analysis.** RNA was extracted from 14-day-old WT and *acinus-2 pinin-1* seedlings using RNeasy mini kit (Qiagen) and treated with TURBO DNA-free Kit (Ambion) to remove any genomic DNA contamination. The mRNA libraries were constructed using NEBNext RNA Library Prep Kit for Illumina following the standard Illumina protocol. Illumina sequencing was performed in the Sequencing Center for Personalized Medicine, Department of Genetics at Stanford University, using an Illumina HiSeq 2000 System. The RNA-seq data have been deposited at the NCBI Gene Expression Omnibus (GEO) database under the accession number GSE110923.

Differential gene expression was analyzed using STAR and Deseq2. Trimmed and quality control-filtered sequence reads were mapped to the *Arabidopsis* reference genome (TAIR10) using STAR (v.2.54) in two pass mode (parameters: −outFilterScoreMinOverLread 0.3, −outFilterMatchNminOverLread 0.3, −outSAMstrandField intronMotif, −outFilterType BySJout, −outFilterIntronMotifs RemoveNoncanonical, −quantMode TranscriptomeSAM GeneCounts)[69]. To obtain uniquely mapping reads, these were filtered by mapping quality (q20), and PCR duplicates were removed using Samtools rmdup (v.1.3.1). Gene expression was analyzed in R (v.3.4.1) using DEseq2 (v.1.16.1)[70]. Significant differentially expressed genes are selected based on adj *p*-value <0.02 and fold change >2.

AS analysis was performed with RACKJ using default setting (online manual available at http://rackj.sourceforge.net/)[71]. Raw intron retention data were analyzed and filtered to reduce false positives with two criteria: (1) fold change of intron retention >2, *p*-value <0.05 in a two-tailed *t*-test and (2) intron RPKM >1 and estimated percentage of IR >5% in the sample that shows increased IR in the intron. Raw exon skipping (ES) data were analyzed and filtered with two criteria: (1) fold change of ES rate >2, *p*-value <0.05 in a two-tailed *t*-test, and (2) increased ES event is supported by reads with RPKM >1 and ES rate >5%. For alternative donor/acceptor usage discovery, only events that appear significantly different in each pair-wise comparison between WT and *acinus-2 pinin-1* (Fisher's exact test *p*-value <0.05) were considered significant and were further filtered with two criteria: (1) fold change >2 and (2) increased alternative donor/acceptor usage is supported by reads with RPKM >1 and rate >5%.

**RNA extraction, reverse transcription PCR.** RNA was extracted from seedlings using Spectrum™ Plant Total RNA Kit (Sigma) and treated with TURBO DNA-free Kit (Ambion) to remove any genomic DNA contaminants. Purified RNA (500 ng) is subjected to cDNA synthesis using RevertAid Reverse Transcriptase (Thermo) with Oligo(dT)$_{18}$ primer. The synthesized cDNA was used for PCR and qPCR analyses. PCR products were analyzed by gel electrophoresis and the PCR band intensities were quantified using ImageJ. The qPCR analyses were performed with the SensiMix™ SYBR® & Fluorescein Kit (Bioline) on a LightCycler 480 (Roches). For each sample, two technical replicates were performed. The comparative cycle threshold method was used for calculating transcript level. Primers used for *FLC* antisense analysis are the same as in the previous publication[37]. Sequences of oligo primers are listed in Supplementary Data 4.

**RNA immunoprecipitation.** RNA-IP was performed using a protocol modified based on published procedures[22]. Briefly, 3 g of tissues of 7-day-old *35S::AtACINUS-GFP/acinus-2* and *35S::GFP* seedlings were cross-linked with 1% (v/v) formaldehyde for 15 min. Cross-linked RNA–protein complexes were extracted in NLB buffer (20 mmol/L Tris-HCl, pH 8.0, 150 mmol/L NaCl, 2 mmol/L EDTA, 1% (v/v) Triton X-100, 0.1% (w/v) SDS, 1 mmol/L PMSF, and 2× Protease Inhibitor (Roche)) and sheared by sonication (25% amplitude, 0.5 s on/0.5 s off for 2 min × 3 cycles on a Branson Digital Sonifier). Immunoprecipitation was carried out with Protein A magnetic beads (Thermo Fisher) that were pre-incubated overnight with homemade anti-GFP antibody (5 μg for each sample) for 1 h on a rotator. Beads were washed five times with 1 mL of NLB buffer (no SDS, 0.5% (v/v) Triton X-100) with 80 U/mL RNase inhibitor. To elute the immuno-complex, 100 μL of elution buffer (20 mmol/L Tris-HCl, pH 8.0, 10 mmol/L EDTA, 1% (w/v) SDS, 800 U/mL RNase inhibitor) was added to the beads and incubated at 65 °C for 15 min. The elute was incubated with 1 μL of 20 mg/mL Protease K at 65 °C for 1 h for protein digestion and reverse-crosslinking. RNA was purified and concentrated using the RNA Clean & Concentrator™ kit (Zymo). On-column DNase digestion was performed to remove DNA contaminations. Samples were kept on ice whenever possible during the experiment. Three biological replicates were performed and the co-immunoprecipitated RNAs were quantified with RT-qPCR, and the results were normalized to *PP2a* and *25S rRNA*[72].

**ChIP-PCR.** ChIP analysis was performed using a similar protocol to the previous publications[73]. Briefly, tissue crosslinking, protein extraction, and immunoprecipitation were carried out as described above for RNA-IP. The beads were washed with low-salt buffer (50 mmol/L Tris-HCl at pH 8.0, 2 mmol/L EDTA, 150 mmol/L NaCl, and 0.5% (v/v) Triton X-100), high-salt buffer (50 mmol/L Tris-HCl at pH 8.0, 2 mmol/L EDTA, 500 mmol/L NaCl, and 0.5% (v/v) Triton X-100), LiCl buffer (10 mmol/L Tris-HCl at pH 8.0, 1 mmol/L EDTA, 0.25 mol/L LiCl, 0.5% (w/v) NP-40, and 0.5% (w/v) sodium deoxycholate), and TE buffer (10 mmol/L Tris-HCl at pH 8.0 and 1 mmol/L EDTA), and eluted with elution buffer (1% (w/v) SDS and 0.1 mmol/L NaHCO₃). After reverse cross-linking and proteinase K digestion, the DNA was purified with a PCR purification kit (Thermo Fisher) and analyzed by PCR. Three biological replicates were performed. *FLC* primers were based on the previous publications[47].

**SILIA-MS quantitative analysis of the AtACINUS interactome.** Stable-isotope-labeling in *Arabidopsis* mass spectrometry (SILIA-MS) was used for quantitative analysis of the AtACINUS interactome. The WT and *acinus-2* plants were grown for 2 weeks at 21 °C under constant light on vertical plates of ¹⁴N or ¹⁵N medium

(Hogland's No. 2 salt mixture without nitrogen 1.34 g/L, 6 g/L Phytoblend, 2 μmol/L propiconazole, and 1 g/L $KNO_3$ or 1 g/L $K^{15}NO_3$ (Cambridge Isotope Laboratories), pH 5.8). About 5 g of tissue was harvested for each sample, ground in liquid nitrogen and stored at −80 °C. Immunoprecipitation was performed as described previously with slight modifications[74]. Briefly, proteins were extracted in 10 mL of MOPS buffer (100 mmol/L MOPS, pH 7.6, 150 mmol/L NaCl, 1% (v/v) Triton X-100, 1 mmol/L phenylmethylsulfonyl fluoride (PMSF), 2× Complete protease inhibitor cocktail, and PhosStop cocktail (Roche)), centrifuged, and filtered through two layers of Miracloth. The flow through was incubated with 20 μg of anti-AtACINUS antibody for 1 h at 4 °C, then 50 μL of protein A agarose beads were added and incubated for another hour, followed by four 2-min washes with immunoprecipitation buffer. At the last wash, $^{14}$N-labeled WT and $^{15}$N-labeled acinus-2 IP samples or reciprocal $^{15}$N-labeled WT and $^{14}$N-labeled acinus-2 IP samples were mixed, and eluted with 2× SDS buffer. The eluted proteins were separated by SDS-PAGE. After Coomassie Brillant blue staining, the whole lane of protein samples was excised in ten segments and subjected to in-gel digestion with trypsin.

The peptide mixtures were desalted using C18 ZipTips (Millipore) and analyzed on an LTQ-Orbitrap Velos mass spectrometer (Thermo Fisher), equipped with a NanoAcquity liquid chromatography system (Waters). Peptides were loaded onto a trapping column (NanoAcquity UPLC 180 μm × 20 mm; Waters) and then washed with 0.1% (v/v) formic acid. The analytical column was a BEH130 C18 100 μm × 100 mm (Waters). The flow rate was 600 nL/min. Peptides were eluted by a gradient from 2–30% solvent B (100% (v/v) acetonitrile/0.1% (v/v) formic acid) over 34 min, followed by a short wash at 50% solvent B. After a precursor scan was measured in the Orbitrap by scanning from mass-to-charge ratio 350 to 1500, the six most intense multiply charged precursors were selected for collision-induced dissociation in the linear ion trap.

Tandem mass spectrometry peak lists were extracted using an in-house script PAVA, and data were searched using Protein Prospector against the Arabidopsis Information Resource (TAIR10) database, to which reverse sequence versions were concatenated (a total of 35,386 entries) to allow estimation of a false discovery rate (FDR). Carbamidomethylcysteine was searched as a fixed modification and oxidation of methionine and N-terminal acetylation as variable modifications. Data were searched with a 10 p.p.m. tolerance for precursor ions and 0.6 Da for fragment ions. Peptide and protein FDRs were set as 0.01 and 0.05. $^{15}$N-labeled amino acids were also searched as a fixed modification for $^{15}$N data. $^{15}$N labeling efficiency was calculated as about 96%, by manually comparing experimental peak envelope data of the $^{15}$N-labeled peptide from top 10 proteins in the raw data to theoretical isotope distributions using Software Protein-prospector (MS-Isotope app). Quantification was done using Protein Prospector which automatically adjusts the L/H ratio with labeling efficiency. The SILIA ratio (WT/acinus-2) was normalized using the average ratios of non-specific interactor ribosomal proteins (with more than five peptides). $^{15}$N labeling samples in general have lower identification rates of proteins because of incomplete (96%) labeling efficiency. The data have been deposited to PRIDE with project accession: PXD020700.

**Label-free mass spectrometric analysis of AtACINUS and its interactome**. The AtACINUS-GFP/acinus-2 and TAP-GFP seedlings[38] were grown for 7 days at 21 °C under constant light on ½ MS medium. Tissues were harvested, ground in liquid nitrogen, and stored at −80 °C.

Immunoprecipitation was performed as described previously with slight modifications[74]. Briefly, proteins were extracted in MOPS buffer (100 mmol/L MOPS, pH 7.6, 150 mmol/L NaCl, 1% (v/v) Triton X-100, 1 mmol/L PMSF, 2× Complete protease inhibitor cocktail, and PhosStop cocktail (Roche)) and 20 μmol/L PUGNAc inhibitor (Sigma), centrifuged, and filtered through two layers of Miracloth, then incubated with a modified version of LaG16-LaG2 anti-GFP nanobody[75] conjugated to Dynabeads (Invitrogen), for 3 h at 4 °C, followed by four 2-min washes with immunoprecipitation buffer and eluted with 2% (w/v) SDS buffer containing 10 mmol/L tris(2-carboxyethyl) phosphine (TCEP) and 40 mmol/L chloroacetamide at 95 °C for 5 min. The eluted proteins were separated by SDS-PAGE. After Colloidal blue staining, the whole lane of protein samples was excised in two segments and subjected to in-gel digestion with trypsin. Three biological experiments were performed.

The peptide mixtures were desalted using C18 ZipTips (Millipore) and analyzed on a Q-Exactive HF hybrid quadrupole-Orbitrap mass spectrometer (Thermo Fisher) equipped with an Easy LC 1200 UPLC liquid chromatography system (Thermo Fisher). Peptides were separated using analytical column ES803 (Thermo Fisher). The flow rate was 300 nL/min and a 120-min gradient was used. Peptides were eluted by a gradient from 3 to 28% solvent B (80% (v/v) acetonitrile/0.1% (v/v) formic acid) over 100 min and from 28 to 44% solvent B over 20 min, followed by a short wash at 90% solvent B. Precursor scan was from mass-to-charge ratio (m/z) 375 to 1600 and top 20 most intense multiply charged precursors were selected for fragmentation. Peptides were fragmented with higher-energy collision dissociation (HCD) with normalized collision energy (NCE) 27.

The raw data were processed by MaxQuant using most of the preconfigured settings[76]. The search was against the same TAIR database as mentioned above. Carbamidomethylcysteine was searched as a fixed modification and oxidation of methionine and N-terminal acetylation as variable modifications. Data were searched with a 4.5 p.p.m. tolerance for precursor ion and 20 p.p.m. for fragment

ions. The second peptide feature was enabled. A maximum of two missed cleavages was allowed. Peptide and protein FDRs were set as 0.01. Minimum required peptide length was seven amino acids. Multiplicity was set to 1. Label-free quantification (LFQ) was enabled. The match between runs option was enabled with a match time window of 0.7 min and an alignment time window of 20 min. Quantification was done on unique and razor peptides and a minimum ratio count was set to 2.

The proteinGroups.txt file generated by MaxQuant were loaded to Perseus[77]. The results were filtered by removing identified proteins by only modified sites, or hits to reverse database and contaminants. LFQ intensity values were logarithmized. The pull-downs were divided to AtACINUS-GFP and TAP-GFP control. Samples were grouped in triplicates and identifications were filtered for proteins having at least three values in at least one replicate group. Signals that were originally zero were imputed with random numbers from a normal distribution (width 0.3, shift = 1.8). Volcano plot was performed with x axis representing the logarithmic ratios of protein intensities between AtACINUS-GFP and TAP-GFP. The hyperbolic curve that separates AtACINUS specific interactor from background was drawn using threshold value FDR 0.01 and curve bend S0 value 2.

LFQ data and SILIA data were combined and filtered to get a high-confidence list of interactors: (1) Significant enrichment in LFQ three biological replicates (FDR = 0.01, S0 = 2); (2) Enrichment of over twofolds in both SILIA biological experiment; or over twofolds in one SILIA experiment, but not identified in second SILIA experiment. If the proteins are only identified and quantified by LFQ three biological replicates, then a higher stringency cut off (enrichment >16-fold, t-value >4) is used. The data were deposited to PRIDE with project accession: PXD020748.

For affinity purification of AtACINUS using in vivo biotinylation, the acinus pinin mutant was transformed with a T-DNA construct that expresses AtACINUS as a fusion with TurboID from the 35S promoter[65]. The AtACINUS-YFP-TurboID/acinus-2 pinin-1 seedlings were treated with 0 or 50 μmol/L biotin for 3 h. The AtACINUS-YFP-Turbo protein was affinity purified using streptavidin beads as previously described[65] using a modified extraction buffer containing 20 μmol/L PUGNAC and 1× PhosphoStop. After on-bead tryptic digestion, the samples were analyzed as described above in the label-free IP-MS section on a Q-Exactive HF instrument. Data were searched as described above but allowing additional modifications: O-GlcNAcylation modification on S/T and neutral loss, O-fucosylation on S/T and neutral loss, phosphorylation on S/T and biotinylation on lysine. The data were deposited to PRIDE with accession number: PXD020749.

**Targeted quantification comparing WT and the acinus-2 pinin-1 double mutant**. The WT and acinus-2 pinin-1 plants were grown on Hoagland medium containing $^{14}$N or $^{15}$N (1.34 g/L Hogland's No. 2 salt mixture without nitrogen, 6 g/L Phytoblend, and 1 g/L $KNO_3$ or 1 g/L $K^{15}NO_3$ (Cambridge Isotope Laboratories), pH 5.8). Proteins were extracted from six samples (one $^{14}$N-labeled Col, two of $^{15}$N-labeled Col, two of $^{14}$N-labeled acinus-2 pinin-1, and one $^{15}$N-labeled acinus-2 pinin-1) individually using SDS sample buffer and mixed as the following: one forward sample F1 ($^{14}$N Col/$^{15}$N acinus-2 pinin-1) and two reverse samples R2 and R3 ($^{14}$N acinus-2 pinin-1/$^{15}$N Col) and separated by the SDS-PAGE gel with a very short run. Two segments (upper part (U) ranging from the loading well to ~50 KD; lower part (L) ranging from ~50 KD to the dye front) were excised, trypsin digested, and analyzed by liquid chromatography mass spectrometry (LC-MS) as described above in the label-free IP-MS section on a Q-Exactive HF instrument using an ES803A analytical column. Data-dependent acquisition was used first to get the peptide information from multiple proteins with peptide mass/charge (m/z), retention time, and MS2 fragments. PININ peptide information was from an IP-MS experiment. For targeted analysis, parallel reaction monitoring (PRM) acquisition[78] using a 20-min window was scheduled with an orbitrap resolution at 60,000, AGC value 2e5, and maximum fill time of 200 ms. The isolation window for each precursor was set at 1.4 m/z unit. Data processing was similar to the previous report[79] with a 5-p.p.m. window using skyline for $^{14}$N- and $^{15}$N-labeled samples. Peak areas of fragments were calculated from each sample, the sum of peak areas from the upper gel segment and the lower gel segment was used to calculate the acinus-2 pinin-1/Col ratios for each peptide, normalized to TUBULIN2 to get the normalized ratios. The median number of multiple ratio measurements is used for each protein.

**Statistics and reproducibility**. Figure 4b, d and Supplementary Fig. 16 show representative results from two independent experiments, each with three biological repeats. Figure 6 and Supplementary Fig. 3a, b show representative results from two independent experiments. Figure 5c shows results from one experiment with three biological repeats.

**Reporting summary**. Further information on research design is available in the Nature Research Reporting Summary linked to this article.

## Data availability

Source data are provided as a supplementary file. Proteomic Data that support the findings of this study have been deposited in Proteomics Identification Database (PRIDE) with the accession codes: PXD020700, PXD020748, PXD020749. The RNA-seq data that support the findings of this study have been deposited in the National Center

for Biotechnology Information Gene Expression Omnibus and are accessible through the GEO series accession number GSE110923. All other related data are available from the corresponding authors upon request.

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

## Acknowledgements

We thank Jeffrey Mugridge and John D Gross for technical assistance. We thank Brian Chait and Michael Rout labs for the LaG16-LaG2 construct and thank Dr. Jixian Zhai for providing the *Arabidopsis* pts intron dataset. We also thank the Salk Institute and *Arabidopsis* Biological Resource Center for the *Arabidopsis* T-DNA insertion lines and Dr. Neil Olszewski for sharing biological materials. This work was supported by National Institutes of Health (NIH) (5R01GM066258 to Z.-Y.W., R01GM135706 to S.L.X., 8P41GM103481 to A.L.B., and 2R01GM047475 to P.H.Q) and the Carnegie endowment fund to the Carnegie mass spectrometry facility.

## Author contributions

Z.D., K.L., J.O., and A.L.B. identified AtACINUS; Z.D., S.L.X., and S.P. analyzed the acinus mutant; Y.B., S.L.X., and D.S. characterized the acinus pinin double mutants; Z.Z. identified the *spy-t1* mutant. Y.B. performed the RNA-seq and RT-PCR analyses. T.H. helped with RNA-seq data analysis; W.N. performed the proteomic analysis of AtACINUS interactome under supervision by A.L.B., P.H.Q. and S.L.X. R.S. performed the targeted mass spectrometry quantification, S.H. performed the affinity purification of biotinylated protein and R.S. prepared the spectra. Z.-Y.W. and S.L.X. conceived the projects; Y.B., S.L.X., and Z.-Y.W. wrote the manuscript.

## Competing interests

The authors declare no competing interests.
