## [Peer Review File · Nature Communications]

Reviewers' comments:

Reviewer #1 (Remarks to the Author):

In a previous publication, the authors found ACINUS to be O-GlcNAcylated. ACINUS is a component of the ASAP complex. ASAP is involved in epigenetic regulation of gene expression, but also splicing. *acinus* and *pinin* mutants are indistinguishable from WT. The authors generated *acinus pinin* double mutants, which exhibited pleiotropic growth defects, altered ABA responses and delayed flowering. The authors used this mutant for transcriptome analysis, in which they found unspliced mRNAs. They further demonstrate that ACINUS interacts with some mRNAs and associates with the FLC gene. In addition, they characterized the ACINUS complex by IP-MS analysis. This manuscript contains lots of interesting and valuable information, which makes it a good fit for the journal. But in its present form, the manuscript also contains some logical inconsistencies. I hope that my comments and suggestions help to remove these inconsistencies before publication:

The authors start with the fact that ACINUS is O-GlcNAcylated. But because ACINUS acts redundantly with PININ, they focused extensively on the *acinus pinin* double mutant. However, this reviewer misses information about the O-GlcNAcylation status of PININ. The rationale of the experiments and the logical flow of the manuscript (e.g. comparison of *acinus pinin* with *spy* and *sec* mutants) require demonstration of O-GlcNAcylation of PININ, too.

SPY and SEC are responsible for O-GlcNAcylation. A weak point of the manuscript is that the authors did not compare the transcriptome of a *sec* and *spy* mutant with that of a *acinus pinin* double. This would have allowed to globally compare splicing defects. Instead, the authors compared only 4 selected splicing events. The results of this experiments are not conclusive (splicing events are either enhanced or repressed) and thus do not allow to establish any functional relationship between O-GlcNAcylation and ACINUS/PININ.

The authors applied an algorithm that detects intron retention events. There are some software tools, which would also allow to detect other splicing defects (such as exon skipping). Because this information might be very useful to understand the function of ACINUS and PININ, this reviewer suggests to reanalyze data with a different software tool. This might also help to compare the effects of the ACINUS/PININ and SPY/SEC in splicing regulation.

The authors suggest that ABH1 might be responsible for the ABA-related defects in *acinus pinin* double mutants. Indeed, a misspliced, and presumably non-functional, isoform of ABH1 accumulates in *acinus pinin* mutants. However, 50% of the ABH1 mRNA in *acinus pinin* mutants is still wild-type mRNA. How does that fit the observation that *abh1* mutants are recessive; thus ABH1 is fully functional when present only at 50% of the WT level?

Figure 5C: The authors show that ACINUS binds to the FLC locus. It would be preferably to perform a quantitative analysis here. In cases where the PCR did not work (e.g. FLC G): Is it actually true that ACINUS does not bind this region? Or did simply the PCR fail? A positive control (input) is missing here. Why did the authors use WT as a negative control?

The *acinus* mutant is not a null allele (see Figure S5). Have the authors generated a true null allele of ACINUS using CRISPR?

Please include the recently published paper by Chen et al., which describes the ASAP complex based on SR45 complex purifications.

Reviewer #2 (Remarks to the Author):

A previous proteomics study in Dr. Wang's lab has identified AtACINUS as an O-GlcNAc modified protein. This manuscript provided genetic and molecular evidence to support the model that AtACINUS was associated with other ASAP complex components and the chromatin remodelers at FLC locus (this merely confirmed findings from Caroline Dean's lab in 2016, Science 353:485-488, but a few additional associated proteins were identified), and AtACINUS was associated with spliceosome components, and transcripts of ABH1 and HAB1, two negative regulators of ABA (this has not been reported before). AtACINUS has a distance paralog AtPININ. Although the acinus single mutant does not have any obvious genotype, the acinus pinin double mutant showed an increase in FLC expression and differential alternative splicing of ABH1 and HAB1, which were in agreement with the double mutant phenotypes - delayed flowering and hypersensitivity to ABA.

This study provided information on an interesting protein complex ASAP and its paralog PSAP. Most previous studies were focused on one component of these complexes, but not the complex as a whole. Since ASAP and PSAP are involved in different levels of gene regulation, this topic would be of interest to others in the research community to understand the complexity in gene regulation beyond just differential gene expression. However, functional distinctions between ASAP and PSAP are unknown in both animal and plant fields, and there isn't a cohesive understanding in terms of how they work in plants.

The evidence provided in this study is mostly convincing to support observed ACINUS's effect on target genes. However, it can be strengthened in the following areas:

1. Introduction does not provide up-to-date background research related to ASAP complex. According to TAIR, there are 33 papers listed for SR45, 9 papers listed for AtSAP18, 3 papers listed for AtACINUS, and 3 papers listed for AtPININ. Ruling out the ones that are cross-listed and the ones that are omics studies non-specific for these three genes, there are still quite a few unique papers published on these ASAP complex components. The authors ignored all of these studies in introduction. Besides, the last sentence of the first paragraph (page 4, lines 83-84) basically ignored Caroline Dean's work published in Science 2016 which they later referenced in Results and Discussion.
2. The Salk line ID for acinus-1 mutant cannot be found in ABRC stocks, likely a typo. There are other typos in the manuscript that need to be corrected.
3. It is not clear if ACINUS's role in alternative splicing is ASAP-dependent or solely due to ACINUS itself. Although the authors very briefly mentioned the overlapping observations in sr45-1 mutant, higher order mutants could be created to address this question better. In addition, SR45 affects AtSAP18 and AtACINUS at the protein level (FIPS, 2019. DOI: fpls.2019.01116). It would be helpful to know the status of SR45 and AtSAP18 in acinus single mutants or ap double mutant to understand the impact of acinus mutation on ASAP complex. A linear association of ACINUS with tested gene targets only gives a very simplistic view of the actual mechanism.
4. Statistical analysis used in this manuscript is insufficient. There was mentioning of statistical tests for only large datasets and supplemental figure 8. No statistical tests were mentioned in other experiments where different groups are compared. Especially in Figure 4c where the authors compared the ratio of ABH1.2/ABH1 with very marginal difference between wild type and single mutants (page 7, lines 192-195). Are these differences statistically significant? If not, this cannot be determined as an ACINUS or PININ-dependent alternative splicing event. Since SEM was used as error bars, a visual check suggests a large p value. But a formal statistical test will be able to tell.
5. Did authors use 12-day seedlings for all CHIP, RIP, PCR and interactome studies? It was mentioned in some places, but not specified in all.

Reviewer #3 (Remarks to the Author):

This study by Yang et al. focuses on two Arabidopsis proteins, AtAcinus and AtPinin, that share an RNPS1-SAP18 binding (RSB) domain and undergo posttranslational O-GlcNAc modification. Single

AtAcinus and AtPinin loss-of-function mutants do not show strong morphological changes, but the double mutant displays severe developmental phenotypes, such as delayed flowering, as well as altered abscisic acid (ABA) responses during seed germination. Large-scale RNA-seq analysis of the double mutant identified many genes with altered gene expression and several with splicing defects. The ABA germination phenotype was linked to defective alternative splicing of the HAB1 protein phosphatase and the ABH1 (CBP80) splicing factor, both key players in the ABA pathway, to which RNA the authors show AtAcinus binds directly. On the other hand, the flowering phenotype was attributed to altered expression of FLC, at which locus AtAcinus is found capable of binding chromatin. Finally, the AtAcinus protein is shown to be O-GlcNAc-tylated and to interact with spliceosomal subunits, EJC related proteins as well as factors from the NTC/NTR and ASAP/PSAP complex. Some work on two O-GlcNAc-transferases, SPINDLY (SPY) and SECRET AGENT (SEC), is also conducted, raising a possible connection between this posttranslational modification and protein function.

The manuscript is well written, and many different experimental approaches are employed to study the AtAcinus Atpinin double mutant, with subsequent focus on AtAcinus. These two Arabidopsis genes have not been studied in detail, and the present work demonstrates their importance in fundamental biological processes. AtAcinus is also shown to integrate the evolutionarily conserved ASAP/PSAP complex, fulfilling a dual function in transcription and RNA splicing. The weakest aspect of the study concerns the functional relevance of O-GlcNAc modification and the roles of SPY and SEC.

Major issues:

1. The authors claim that their study uncovers a function for O-linked sugar modifications in alternative splicing and transcription (e.g. lines 93-97, 295-298, 374-380). This is an overinterpretation of the data, which only show that AtAcinus is O-GlcNAc-tylated and plays a role in transcription and splicing. To address the role of O-linked sugar modifications, the authors perform RT-PCR experiments to study individual alternative splicing events altered in both the O-GlcNAc-transferase mutants *spy* and *sec*. This approach is not entirely convincing nor adequately discussed. For instance, only one event out of 10 is altered in the *sec* mutant when compared to the AtAcinus AtPinin double mutant, but it shows the opposite trend. The same alternative splicing events were tested in the loss of function *spy* mutant, where the detected changes in mRNA isoform abundance were more similar to those in the AtAcinus AtPinin double mutant, and four (instead of one) events out of 10 were changed. Only SEC (not SPY) has been previously shown to possess O-GlcNAc transferase properties, and in some cases the functional relevance this modification has been uncovered (e.g. Zentella et al. 216). Further experimental work would be required to support the claim for a role of O-linked sugar modifications in the studied biological processes, for example through site-directed mutagenesis of the modified amino acid residues and subsequent phenotypical and protein function analyses (e.g. interactions, changes in alternative splicing and/or transcription). Otherwise, the claims and conclusions regarding O-GlcNAc-tylation need to be revised and significantly toned down. The potential role of this posttranslational modification also needs to be further addressed in the discussion section.
2. The FLC chromatin immunoprecipitation experiments require negative control(s) e.g. IgG IP would show the levels of unspecific DNA binding in the absence of the IP antibody and/or PCR analysis of the immunoprecipitated samples for an unrelated gene such as ACTIN would show that the observed enrichment is specific.
3. The authors do not indicate whether the data from the large-scale analysis were made publicly accessible. These should be available.

Minor comments:

1. For clarity, all the abbreviations used should be defined (e.g. line 103 SAP domain, RRM domain).
2. The sentence "Another RSB-containing protein, Pinin, forms a similar protein complex named PSAP" (line 80-81) requires a citation.
3. The RNA-sequencing is stated to have resulted in a minimum of 22.4 million reads per sample. This is relatively low for an in-depth analysis of alternative splicing events. The authors should

consider (and discuss) the possibility that some events have been missed due to this.

4. The alternative gene name for ABH1, CBP80, should be mentioned in the text at least once.

5. Lines 285-288: Five proteins are said to be involved in regulating FLC, but six names are listed. Please revise.

6. Typo in line 331: ("seed germination. f, two most important developmental transitions")

7. In the legend of Figure 1, the names for domain abbreviations should be given.

8. In Figure 1.a, the sites of O-GlcNAc in the protein sequences should be indicated.

9. In Figure 1.b, it would be better to have the sequences of the AtAcinus and AtPinin RSB domains next to each other and the consensus sequence below. Color coding would also help the panel's readability.

10. Figure 5.c would also benefit from color coding of the positive fragments in the gene's schematic diagram.

11. In the description of the germination assays in the Materials and Methods section (page 40), germination is stated to have been assessed by scoring cotyledon emergence. This is likely a mistake. Should be radicle emergence.

Reviewer #4 (Remarks to the Author):

the manuscript „ An O-GlcNAc-modified protein promotes seed germination and flowering through alternative RNA splicing and transcription of key regulators in Arabidopsis“ is postulates that the modification of the Thr79 of AtACINUS with an O-linked sugar affects alternative RNA splicing, that in turn, influences the protein network that integrates nutrient signaling with hormone (ABA) signaling and plant developmental transitions. This aspect is new, that makes the manuscript interesting for other researchers in the field. If these claims will turn to be correct (i.e. properly justified), the molecular mechanisms behind seed germination, seedling development and flowering onset would become much better understood. In this context, the manuscript is interesting, and I would encourage the authors to make the efforts to appropriately improve it to publish finally in Nature Communications. And the main problem on this way would be, to my mind, insufficient justification of the claims at the protein level. Below are some recommendations, which would be useful to improve the manuscript in this sense. The manuscript is clearly written, the language quality is good, the work is perfectly good proof read for typos and spelling mistakes, making, in general, a very good impression.

Major remarks

1. Please, use for molarity "mol/L", but not "M". I know that biologists like this, but "M" – is not SI.

2. The presented strategy for the interactome study is not optimal: it is, on one hand, prone to fals-positives (non-specific binding), on another – dissociation of some partners can occur during the treatment. I would suggest complementing this section with cross-linking experiments. Please, check the concept of in vivo covalent protein cross-linking, applicable for plants (<https://doi.org/10.1002/pmic.201500310>). Adjusting the linker type, you can address the distances between partners.

3. All conclusions about the AtACINUS interactome can be done only on the basis of the data, corrected for multiple comparisons. Only the proteins which fulfill the corrected criteria can be judged as significantly associated with the target protein.

4. The glycosylation part, to my mind, needs to be less speculative. For this, I would suggest two actions: (i) computer modeling of the conformational changes, caused by modifications, i.e. bioinformatics needs to be confirmed with computational chemistry tools, and (ii) using AtACINUS mutants with replacement of Thr79 – this will exclude O-GlcNAcylation and give a good hint to the role of this modification in the function of this protein.

Minor remarks

1. Line 77: please, put "In animals Acinus forms..."

2. Please, when you use "%", define what kind of – v/v or w/v

3. Please, m/z – everywhere in italic.

4. "3 biological repeats" – do you mean "n = 3" or "three biological replicates"

5. Figure 7: The presentation of the spectra is not really informative, as all marks are too small to see. Also, individual fragment ions detectable in the figure can be marked in the sequence, also given at the same panel. This will give the reader an idea about sequence coverage. Then, Volcano plot is also not really needed in the main text. Actually, the information is provided in a supplementary excel table in a much more useful form. However, keep in mind please, that this table does not give any information on correction for the multiple comparisons. I think that Perseus delivers q value. Otherwise, just calculate manually. We are usually using Benjamini-Hochberg correction.

Reviewer #1

In a previous publication, the authors found ACINUS to be O-GlcNAcylated. ACINUS is a component of the ASAP complex. ASAP is involved in epigenetic regulation of gene expression, but also splicing. *acinus* and *pinin* mutants are indistinguishable from WT. The authors generated *acinus pinin* double mutants, which exhibited pleiotropic growth defects, altered ABA responses and delayed flowering. The authors used this mutant for transcriptome analysis, in which they found unspliced mRNAs. They further demonstrate that ACINUS interacts with some mRNAs and associates with the FLC gene. In addition, they characterized the ACINUS complex by IP-MS analysis. This manuscript contains lots of interesting and valuable information, which makes it a good fit for the journal. But in its present form, the manuscript also contains some logical inconsistencies. I hope that my comments and suggestions help to remove these inconsistencies before publication:

1. The authors start with the fact that ACINUS is O-GlcNAcylated. But because ACINUS acts redundantly with PININ, they focused extensively on the acinus pinin double mutant. However, this reviewer misses information about the O-GlcNAcylation status of PININ. The rationale of the experiments and the logical flow of the manuscript (e.g. comparison of acinus pinin with spy and sec mutants) require demonstration of O-GlcNAcylation of PININ, too.

Response: We thank this reviewer for concluding that our work “contains lots of interesting and valuable information, which makes it a good fit for the journal”. We agree that the genetic redundancy suggests that ACINUS and PININ play same or overlapping essential roles under our experimental conditions. However, given their evolutionary distance, it is conceivable that they share some essential functions (such bringing SR45 and SAP18 together in the ASAP and PSAP complex) but also have some distinct functions including response to different signals, which could only be uncovered by analysis of their single mutants under special conditions. We have performed additional MS analysis and detected O-GlcNAcylation and O-fucosylation modifications on AtACINUS. However, we have not been able to detect O-GlcNAcylation or O-fucosylation of PININ. We have performed IP-MS experiments using 35S::YFP-PININ transgenic lines and achieved 58% sequence coverage of PININ, but we did not detect either O-GlcNAcylation or O-fucosylation on PININ. The PININ protein level is lower than ACINUS according to our targeted quantification (supplemental Figure 12 and supplemental table 5) and iBAQ data from recent Arabidopsis proteome paper (Mergner et al., 2020, Nature), making detection of PTMs on PININ more challenging. Therefore, whether AtPININ is O-GlcNAcylated and O-fucosylated remains unknown. However, the phenotypes of altered intron splicing in the *spy* and *sec* mutants provide genetic evidence for these PTMs regulating AtACINUS/AtPININ-dependent RNA splicing. We have added related information into discussion.

2. SPY and SEC are responsible for O-GlcNAcylation. A weak point of the manuscript is that the authors did not compare the transcriptome of a sec and spy mutant with that of a acinus pinin double. This would have allowed to globally compare splicing defects. Instead, the authors compared only 4 selected splicing events. The results of this experiments are not conclusive (splicing events are either enhanced or repressed) and thus to do not allow to establish any functional relationship between O-GlcNAcylation and ACINUS/PININ.

Response: We agree that it would be better if we can globally compare splicing defects. However, this is out of the scope of this manuscript. One reason for not doing this experiment yet is that we need a conditional double *spy sec* mutant to fully understand the functions

including their redundant functions. On the other hand, we believe our RT-PCR experiments are conclusive and does demonstrate a functional relationship between O-glycosylation and AtACINUS/PININ. We compared 10 splicing events and 4 of them were affected by *spy* or *sec*. The complex effects on these 4 splicing events do not suggest a simple relationship of activation or repression of ACINUS/PININ. Instead, the results support an interesting scenario that the sugar modifications have different effects on different RNA substrates. Therefore, our results indicate that the effects of O-GlcNAcylation on ACINUS/PININ that are target gene specific.

We believe our results provide the first genetic evidence for a function of O-glycosylation in regulating RNA splicing.

3. The authors applied an algorithm that detects intron retention events. There are some software tools, which would also allow to detect other splicing defects (such as exon skipping). Because this information might be very useful to understand the function of ACINUS and PININ, this reviewer suggests to reanalyze data with a different software tool. This might also help to compare the effects of the ACINUS/PININ and SPY/SEC in splicing regulation.

Response: The Rack J software does detect exon skipping and alternative splicing site selection. As shown in Supplemental figure 6, we did detect some of these other splicing defects.

The authors suggest that ABH1 might be responsible for the ABA-related defects in *acinus pinin* double mutants. Indeed, a misspliced, and presumably non-functional, isoform of ABH1 accumulates in *acinus pinin* mutants. However, 50% of the ABH1 mRNA in *acinus pinin* mutants is still wild-type mRNA. How does that fit the observation that *abh1* mutants are recessive; thus ABH1 is fully functional when present only at 50% of the WT level?

Response: We concluded that “AS events in ABH1 and HAB1 are likely the major mechanisms by which AtACINUS modulates ABA signaling dynamics and seed germination”. Several lines of evidence support the role of the ACINUS/AtPININ-ABH1 branch. First, we observed a significant correlation of transcriptomic changes between *abh1* mutant and *acinus pinin* mutant (Supplementary Fig 8). Second, RNAi-knockdown of ABH1 RNA to 30% in potato caused obvious phenotypes (Pieczynski et al., 2013, Plant Biotechnology Journal 11, 459–469). Third, ABH1 protein level was decreased in the *sr45* mutant (Chen et al., 2019) and SR45 level is dramatically reduced in the *acinus pinin* mutant (our supplementary Fig 13). Therefore, the reduction of ABH1 RNA level in *acinus pinin* mutant should contribute to the phenotype, though determining the exact amount of this contribution experimentally is not trivial. While loss-of-function mutations of most genes are recessive (gene fully functional when present at 50% WT level), a 2-fold decrease or increase of RNA level is usually considered biological relevant. We acknowledge that our results do not allow us to determine how much ABH1 contributes, relative to HAB1 or other unknown mechanisms, to the phenotype observed in *acinus pinin* double mutant.

Figure 5C: The authors show that ACINUS binds to the FLC locus. It would be preferably to perform a quantitative analysis here. In cases where the PCR did not work (e.g. FLC G): Is it actually true that ACINUS does not bind this region? Or did simply the PCR fail? A positive control (input) is missing here. Why did the authors used WT as a negative control?

Response: We have tested the PCR primers before the ChIP experiment and confirmed that all primers worked. The PCR products from inputs are now shown in the figures.

Using WT (lacking the protein to be immunoprecipitated) as a negative control is better than a control that lacks the antibody because antibodies can have non-specific bindings which cause false positive results. In principle, a transgenic plant expressing GFP alone should be used as the negative control to eliminate any binding due to GFP. In our previous ChIP and IP-MS studies (Sun et al., 2010, *Developmental Cell* 19, 765–777; Bu et al., 2017), we did not observe any difference between 35S-GFP and WT. We hence use WT as a negative control (without antigen) when we do IP using the anti-GFP antibody.

The acinus mutant is not a null allele (see Figure S5). Have the authors generated a true null allele of ACINUS using CRISPR?

Response: We are not sure why this reviewer thought the *acinus* mutant is not a null allele. The T-DNA insertion truncated about 2/3 of the coding sequence in *acinus-2* and about 1/4 in *acinus-1*. The two alleles cause similar phenotypes. While the RNA-seq tracks of PININ look like full-length mRNA was detected, there is a gap at the T-DNA insertion site (marked by the dashed line in Figure S5) and no read spanning the T-DNA insertion site was detected. Therefore, the PININ mRNA was expressed as two separate fragments. We have performed a targeted mass spectrometry analysis (Parallel reaction monitoring) of ACINUS and PININ: ACINUS peptides N-terminal of the T-DNA insertion was detected but at a much reduced level (<20% of WT); No peptides encoded by the C-terminal to the T-DNA insertion site was detectable (signal was very close to noise level) in the *acinus pinin* mutant (supplemental Figure 11 and supplemental table 5). For PININ, both N-terminal and C-terminal are undetectable (Supplemental Figure 12 and supplemental table 5). In both cases, RSB domains are not expressed. We believe the *acinus* and *pinin* mutants are null alleles.

Please include the recently published paper by Chen et al., which describes the ASAP complex based on SR45 complex purifications.

Response: We thank the reviewer for this comment, and we have added this paper in introduction and discussion. However, the paper by Chen et al., (2019) describes the ASAP complex based on only bioinformatic analysis of the protein sequences/domains. Proteomic experiments were performed to quantify the proteins of the *sr45* mutant; no complex purifications were described in this paper.

Reviewer #2

A previous proteomics study in Dr. Wang's lab has identified AtACINUS as an O-GlcNAc modified protein. This manuscript provided genetic and molecular evidence to support the model that AtACINUS was associated with other ASAP complex components and the chromatin remodelers at FLC locus (this merely confirmed findings from Caroline Dean's lab in 2016, *Science* 353:485-488, but a few additional associated proteins were identified), and AtACINUS was associated with spliceosome components, and transcripts of ABH1 and HAB1, two negative regulators of ABA (this has not been reported before). AtACINUS has a distance paralog AtPININ. Although the *acinus* single mutant does not have any obvious genotype, the *acinus pinin* double mutant showed an increase in FLC expression and differential alternative splicing of ABH1 and HAB1, which were in agreement with the double mutant phenotypes - delayed flowering and hypersensitivity to ABA.

This study provided information on an interesting protein complex ASAP and its paralog PSAP. Most previous studies were focused on one component of these complexes, but not the

complex as a whole. Since ASAP and PSAP are involved in different levels of gene regulation, this topic would be of interest to others in the research community to understand the complexity in gene regulation beyond just differential gene expression. However, functional distinctions between ASAP and PSAP are unknown in both animal and plant fields, and there isn't a cohesive understanding in terms of how they work in plants.

The evidence provided in this study is mostly convincing to support observed ACINUS's effect on target genes. However, it can be strengthened in the following areas:

1. *Introduction does not provide up-to-date background research related to ASAP complex. According to TAIR, there are 33 papers listed for SR45, 9 papers listed for AtSAP18, 3 papers listed for AtACINUS, and 3 papers listed for AtPININ. Ruling out the ones that are cross-listed and the ones that are omics studies non-specific for these three genes, there are still quite a few unique papers published on these ASAP complex components. The authors ignored all of these studies in introduction. Besides, the last sentence of the first paragraph (page 4, lines 83-84) basically ignored Caroline Dean's work published in Science 2016 which they later referenced in Results and Discussion.*

Response: We thank the reviewer for these comments, and we have added related references and cited Caroline Dean's paper in the introduction and discussion. We would like to point out that Carolina Dean's paper discovered ACINUS as a protein associated with VAL1 and VAL2, but didn't provide functional analysis of ACINUS. Our work provides critical genetic (mutant phenotypes) and molecular evidence (interaction with *FLC* promoter) for ACINUS function in *FLC* regulation.

2. *The Salk line ID for acinus-1 mutant cannot be found in ABRC stocks, likely a typo. There are other typos in the manuscript that need to be corrected.*

Response: We thank the reviewer for pointing out the typos and we have corrected this typo and other typos.

3. *It is not clear if ACINUS's role in alternative splicing is ASAP-dependent or solely due to ACINUS itself. Although the authors very briefly mentioned the overlapping observations in sr45-1 mutant, higher order mutants could be created to address this question better. In addition, SR45 affects AtSAP18 and AtACINUS at the protein level (FIPS, 2019. DOI: fpls.2019.01116). It would be helpful to know the status of SR45 and AtSAP18 in acinus single mutants or ap double mutant to understand the impact of acinus mutation on ASAP complex. A linear association of ACINUS with tested gene targets only gives a very simplistic view of the actual mechanism.*

Response: Based on review's comments, we have performed a quantitative MS analysis and showed that the levels of SR45 and SAP18 proteins are drastically decreased in the *acinus pinin* mutant, with SR45 decreased to 3.9% and SAP18 to ~2.7% of wild-type level (Supplementary figure 13 and 14). These are more dramatic changes than those observed in the *sr45* mutant (12.8% for AtSAP18 and 66.8% for AtACINUS; Chen et al., 2019 FIPS). These results indicate that AtACINUS and AtPININ (and thus formation of ASAP and PSAP complexes) are required for the accumulation of AtSAP18 and SR45. With such dramatic reduced SR45 level, the *acinus pinin* double mutant is equivalent to the *acinus pinin sr45* triple mutant, and a comparison between them will not reveal SR45-independent functions of AtACINUS/AtPININ.

4. *Statistical analysis used in this manuscript is insufficient. There was mentioning of statistical tests for only large datasets and supplemental figure 8. No statistical tests were mentioned in other experiments where different groups are compared. Especially in Figure 4c where the authors compared the ratio of ABH1.2/ABH1 with very marginal difference between wild type and single mutants (page 7, lines 192-195). Are these differences statistically significant? If not, this cannot be determined as an ACINUS or PININ-dependent alternative splicing event. Since SEM was used as error bars, a visual check suggests a large p value. But a formal statistical test will be able to tell.*

Response: We thank the reviewer for these comments, and we have added statistical analysis in the figures.

5. *Did authors use 12-day seedlings for all CHIP, RIP, PCR and interactome studies? It was mentioned in some places, but not specified in all.*

Response: We have added these experimental details to figure legends as well as in methods.

Reviewer #3

This study by Yang et al. focuses on two Arabidopsis proteins, AtAcinus and AtPinin, that share an RNPS1-SAP18 binding (RSB) domain and undergo posttranslational O-GlcNAc modification. Single AtAcinus and AtPinin loss-of-function mutants do not show strong morphological changes, but the double mutant displays severe developmental phenotypes, such as delayed flowering, as well as altered abscisic acid (ABA) responses during seed germination. Large-scale RNA-seq analysis of the double mutant identified many genes with altered gene expression and several with splicing defects. The ABA germination phenotype was linked to defective alternative splicing of the HAB1 protein phosphatase and the ABH1 (CBP80) splicing factor, both key players in the ABA pathway, to which RNA the authors show AtAcinus binds directly. On the other hand, the flowering phenotype was attributed to altered expression of FLC, at which locus AtAcinus is found capable of binding chromatin. Finally, the AtAcinus protein is shown to be O-GlcNAc-tylated and to interact with spliceosomal subunits, EJC related proteins as well as factors from the NTC/NTR and ASAP/PSAP complex. Some work on two O-GlcNAc-transferases, SPINDLY (SPY) and SECRET AGENT (SEC), is also conducted, raising a possible connection between this posttranslational modification and protein function. The manuscript is well written, and many different experimental approaches are employed to study the AtAcinus Atpinin double mutant, with subsequent focus on AtAcinus. These two Arabidopsis genes have not been studied in detail, and the present work demonstrates their importance in fundamental biological processes. AtAcinus is also shown to integrate the evolutionarily conserved ASAP/PSAP complex, fulfilling a dual function in transcription and RNA splicing. The weakest aspect of the study concerns the functional relevance of O-GlcNAc modification and the roles of SPY and SEC.

Responses: We thank the reviewer's positive comments on the significance and impact of our work.

Major issues:

1. *The authors claim that their study uncovers a function for O-linked sugar modifications in alternative splicing and transcription (e.g. lines 93-97, 295-298, 374-380). This is an overinterpretation of the data, which only show that AtAcinus is O-GlcNAc-tylated and plays a role in transcription and splicing. To address the role of O-linked sugar modifications, the*

*authors perform RT-PCR experiments to study individual alternative splicing events altered in both the O-GlcNAc-transferase mutants *spy* and *sec*. This approach is not entirely convincing nor adequately discussed. For instance, only one event out of 10 is altered in the *sec* mutant when compared to the *AtAcinus AtPinin* double mutant, but it shows the opposite trend. The same alternative splicing events were tested in the loss of function *spy* mutant, where the detected changes in mRNA isoform abundance were more similar to those in the *AtAcinus AtPinin* double mutant, and four (instead of one) events out of 10 were changed.*

Responses: Our novel claim is the function of O-glycosylation in alternative splicing, as the function in transcription has been well established. In addition to *AtACINUS* being O-GlcNAcylated and playing a role in splicing, another key evidence supporting our claim is that a significant fraction of *AtACINUS/AtPININ*-dependent introns are affected in the *spy* and *sec* mutants. The number of AS events affected in *sec* (1 out of 10 *AtACINUS*-dependent splicing events) and *spy* (4 of 10) are consistent with the phenotype severities of the mutants (*sec* mutant is morphologically not distinguishable from WT at adult stage while *spy* shows dramatic phenotypes), stronger effects are expected in the *spy sec* double mutant which is lethal. The opposite effects of *spy* and *sec* on the *TRM4D* intron is consistent with the notion that O-GlcNAc and O-fucose modifications can have opposite effects on some protein functions as demonstrated for the DELLA proteins, whereas the embryo lethal phenotype of *spy sec* double mutant suggests that redundant functions on certain targets. Overall, the functional relationship between SPY and SEC is likely to vary with target proteins and pathways and requires analyses of additional target proteins.

Only SEC (not SPY) has been previously shown to possess O-GlcNAc transferase properties, and in some cases the functional relevance this modification has been uncovered (e.g. Zentella et al. 216). Further experimental work would be required to support the claim for a role of O-linked sugar modifications in the studied biological processes, for example through site-directed mutagenesis of the modified amino acid residues and subsequent phenotypical and protein function analyses (e.g. interactions, changes in alternative splicing and/or transcription). Otherwise, the claims and conclusions regarding O-GlcNAc-tylation need to be revised and significantly toned down. The potential role of this posttranslational modification also needs to be further addressed in the discussion section.

Responses: While we agree that the function of the O-glycosylation of *AtACINUS* need ultimate confirmation by mutagenesis of the modification sites, *spy*'s effects on a high percentage (4 of 10) of *AtACINUS/AtPININ*-dependent introns support the involvement of *AtACINUS* in O-fucose regulation of the splicing of these introns.

We have added discussion of the potential roles of O-GlcNAcylation and O-fucosylation. While we were revising our manuscript, Suzanne Walker's group reported that the inhibitor of OGT causes splicing of post-transcriptionally splice (pts) introns in a human cell line (Pan et al., *Nucleic Acids Res.*, April 2020), and another group reported large numbers of pts introns in *Arabidopsis* and increase of intron retention under stress conditions (Jia et al., *Nature Plants*, July 2020). Several splicing factors involved in pts intron splicing are among the *AtACINUS* interactome, and a large portion of the introns retained in the *acinus pinin* mutant are pts introns. We added discussion of these recent developments and the possibility that *AtACINUS* plays important roles in the splicing of a subset of introns including pts introns downstream of the metabolic signals transduced by SPY/O-fucose and SEC/O-GlcNAc.

2. *The FLC chromatin immunoprecipitation experiments require negative control(s) e.g. IgG IP would show the levels of unspecific DNA binding in the absence of the IP antibody and/or PCR*

analysis of the immunoprecipitated samples for an unrelated gene such as ACTIN would show that the observed enrichment is specific.

Response: We used plants without the antigen as our negative control, instead of a negative control lacking the antibody. We believe our control is better because it controls for non-specific binding by the antibody and beads. We use CNX5 as a reference gene (Non-target) to show the background DNA levels.

3. The authors do not indicate whether the data from the large-scale analysis were made publicly accessible. These should be available.

Response: We have deposited proteomic data to PRIDE and RNA-seq data to NCBI Gene Expression Omnibus (GEO) database and included project accession numbers in the method and materials.

Minor comments:

1. For clarity, all the abbreviations used should be defined (e.g. line 103 SAP domain, RRM domain).

Response: We have added full domain names in the text

2. The sentence “Another RSB-containing protein, Pinin, forms a similar protein complex named PSAP” (line 80-81) requires a citation.

Response: We have added the citation.

3. The RNA-sequencing is stated to have resulted in a minimum of 22.4 million reads per sample. This is relatively low for an in-depth analysis of alternative splicing events. The authors should consider (and discuss) the possibility that some events have been missed due to this.

Response: We thank the reviewer for this comment, and we have added discussion that further in-depth RNA-seq could improve our understanding of AtACINUS functions in AS.

4. The alternative gene name for ABH1, CBP80, should be mentioned in the text at least once.

Response: We have added CBP80 into the text

5. Lines 285-288: Five proteins are said to be involved in regulating FLC, but six names are listed. Please revise.

Response: Only five protein names are mentioned.

6. Typo in line 331: (“seed germination. f, two most important developmental transitions”)

Response: We have corrected the typo.

7. In the legend of Figure 1, the names for domain abbreviations should be given.

Response: We have added full domain names to the legend

8. In Figure 1.a, the sites of O-GlcNAc in the protein sequences should be indicated.

Response: We have indicated position of O-GlcNAc and O-Fuc modifications in the protein sequence.

9. In Figure 1.b, it would be better to have the sequences of the AtAcinus and AtPinin RSB domains next to each other and the consensus sequence below. Color coding would also help the panel's readability.

Response: We have moved sequences of the AtAcinus and AtPinin RSB domains next to each other and moved the consensus sequence below. Consensus amino acids are now highlighted in green.

10. Figure 5.c would also benefit from color coding of the positive fragments in the gene's schematic diagram.

Response: We have highlighted positive fragments in blue.

11. In the description of the germination assays in the Materials and Methods section (page 40), germination is stated to have been assessed by scoring cotyledon emergence. This is likely a mistake. Should be radicle emergence.

Response: We have corrected the mistake.

Reviewer #4 (Remarks to the Author):

the manuscript „ An O-GlcNAc-modified protein promotes seed germination and flowering through alternative RNA splicing and transcription of key regulators in Arabidopsis“ is postulates that the modification of the Thr79 of AtACINUS with an O-linked sugar affects alternative RNA splicing, that in turn, influences the protein network that integrates nutrient signaling with hormone (ABA) signaling and plant developmental transitions. This aspect is new, that makes the manuscript interesting for other researchers in the field. If these claims will turn to be correct (i.e. properly justified), the molecular mechanisms behind seed germination, seedling development and flowering onset would become much better understood. In this context, the manuscript is interesting, and I would encourage the authors to make the efforts to appropriately improve it to publish finally in Nature Communications. And the main problem on this way would be, to my mind, insufficient justification of the claims at the protein level. Below are some recommendations, which would be useful to improve the manuscript in this sense. The manuscript is clearly written, the language quality is good, the work is perfectly good proof read for typos and spelling mistakes, making, in general, a very good impression.

Major remarks

1. Please, use for molarity “mol/L”, but not “M”. I know that biologists like this, but “M” – is not SI.

Response: We thank the reviewer for the positive comments on the significance and impact of our work. We have changed M to mol/L, though we see M used in other Nat Comm papers.

2. The presented strategy for the interactome study is not optimal: it is, on one hand, prone to false-positives (non-specific binding), on another – dissociation of some partners can occur during the treatment. I would suggest complementing this section with cross-linking experiments. Please, check the concept of *in vivo* covalent protein cross-linking, applicable for plants (<https://doi.org/10.1002/pmic.201500310>). Adjusting the linker type, you can address the distances between partners.

Response: We believe our interactome experiments went beyond common practice of the field, and our data is of high confidence and contain few, if any, false positives. We have performed two IP-MS experiments, using different anti-GFP antibody and anti-ACINUS antibody. In one IP-MS experiment, we used stable isotope-labeling quantitative mass spectrometry, using *acinus-2* mutant as negative control, with isotope swapped in replicates. In another experiment, label-free quantitation was used with three biological repeats. We believe our interactome analysis is of higher standard than the common practice and our data is of high confidence.

We agree with the reviewer that our experiments would not identify transient interactors that disassociate during the purification process, a common limitation of such experiments. It would be interesting, but beyond the scope of this study, to identify such transient interactors. We have been actively experimenting with cross-linking IP-MS, but so far this approach is not effective in identifying interacting proteins as very few inter-linked peptides can be identified due to low stoichiometric nature of these peptides, an obstacle that is discussed in the reference provided by the reviewer. Nevertheless, our conclusions are based on interactors identified with high confidence, and we do not claim that all interactors have been identified.

3. All conclusions about the AtACINUS interactome can be done only on the basis of the data, corrected for multiple comparisons. Only the proteins which fulfill the corrected criteria can be judged as significantly associated with the target protein.

Response: The statistical tests for label-free experiment, performed with Perseus (FDR=0.01, S0=2), accounted for multiple testing and is used to filter for significant interactors. All of these interactors identified by label-free experiment, except one protein with very high fold enrichment, are reproducibly detected using the metabolic labeling method with at least 2-fold enrichment.

4. The glycosylation part, to my mind, needs to be less speculative. For this, I would suggest two actions: (i) computer modeling of the conformational changes, caused by modifications, i.e. bioinformatics needs to be confirmed with computational chemistry tools, and (ii) using AtACINUS mutants with replacement of Thr79 – this will exclude O-GlcNAcylation and give a good hint to the role of this modification in the function of this protein.

Response: We agree that further experimental work is required to firmly demonstrate the function of O-linked sugar modifications of AtACINUS. We do not have the expertise to perform computer modeling or to evaluate such analysis. Site-directed mutagenesis can provide additional supporting evidence, but is beyond the scope of this study and also has several limitations. First, these sites might be also phosphorylated and thus mutagenesis may not prove the function of O-linked sugar modification. Second, additional sites might be modified, but not detected in our experiments, and thus mutation of one site may not have a strong effect. Such direct proof of the function of O-glycosylation will require extensive effort and is beyond the scope of this study. As discussed above, the effects of *spy* mutation on the splicing of a large portion (4 of 10) of AtACINUS/PININ-dependent introns provide strong genetic evidence supporting the function of O-glycosylation of AtACINUS.

Minor remarks

1. Line 77: please, put “In animals Acinus forms...”

Response: We have corrected the typo.

2. Please, when you use “%”, define what kind of – v/v or w/v

Response: we have defined v/v or w/v as suggested by the reviewer.

3. Please, m/z – everywhere in italic.

Response: We have corrected the format.

4. “3 biological repeats” – do you mean “n = 3” or “three biological replicates”

Response: We have changed to three biological replicates.

5. Figure 7: The presentation of the spectra is not really informative, as all marks are too small to see. Also, individual fragment ions detectable in the figure can be marked in the sequence, also given at the same panel. This will give the reader an idea about sequence coverage. Then, Volcano plot is also not really needed in the main text. Actually, the information is provided in a supplementary excel table in a much more useful form. However, keep in mind please, that this table does not give any information on correction for the multiple comparisons. I think that Perseus delivers q value. Otherwise, just calculate manually. We are usually using Benjamini-Hochberg correction.

Response: We have enlarged the spectra to show more details. We have also marked individual fragment ions. We have moved the volcano plot to supplementary figure. We filtered significantly enriched interactors from the label-free quantification experiment by setting the parameter S0=2, as a common practice used by most of the Maxquant/Perseus analysis. This type of setting is slightly less stringent on proteins with greater folder changes (more biologically

meaningful) because at nonzero S_0 the difference of means play a role in addition to the p value (Tusher, Tibshirani and Chu(2001), PNAS 98, PP5116-21). We also provided the Perseus-calculated Permutation-based and Benjamini-Hochberg-based q values in the excel tables as requested by the reviewer.

REVIEWERS' COMMENTS

Reviewer #2 (Remarks to the Author):

In this revision, the authors have address most of the previous concerns. The most interesting new information is the striking decrease of SR45 and SAP18 protein levels in the double mutants. This make a stronger argument for an impact of ASAP/PASAP in the double mutants at the tested developmental stage. However, molecular interactions are temporally and spatially regulated in a dynamic fashion. A direct comparison between a vegetative stage setup in this manuscript with the results from inflorescence may or may not reflect the actual molecular dynamics regardless of developmental control. For example, SR45 is expressed at a much lower level in vegetative tissues than inflorescence (www.plantphysiol.org/cgi/doi/10.1104/pp.109.138180). It would not be a surprise if SAP18 exhibited a similar expression pattern. It is unclear if ACINUS and PININ would follow a similar expression pattern since the authors used overexpression lines in the experiment. Although it seems not directly relevant, it could be a reason for the difference in SAP18 protein level between this work and the report in inflorescence before if simply considering stoichiometry.

One correction that the authors did not make is the error in mutant line info. SALK_078854 is an insertion in gene AT3G25750, not ACINUS. The T-DNA line name for pinin mutant can not be found in the ABRC database either. It is an obvious concern if a paper publishes misleading information on the genetic materials it used.

Reviewer #3 (Remarks to the Author):

The authors have conducted a significant amount of additional work in this revision that improves the manuscript and addresses the concerns raised by this reviewer. Minor editing is still required but should be easy to solve.

Reviewer #4 (Remarks to the Author):

after reading the answers of the authors and the revized manuscript, I have the following remarks.

1. It is very nice that the authors their experiment convincing. Yes, they good a good job, but it needs to be understood that IP-MS is not really state of the art. Therefore, two remarks need to be done: 1) the authors need to mark clearly that unknown number of interaction partners could be lost during handling; 2) some of 45 proteins representing the interactome could be precipitated via secondary, not native, interactions. Without these remarks the manuscript should not be accepted. Or – use doubt-free techniques.

2. About the answer to my critique point # 4: the authors write: "As discussed above, the effects of spy mutation on the splicing of a large portion (4 of 10) of AtACINUS/PININ-dependent introns provide strong genetic evidence supporting the function of O-glycosylation of AtACINUS.". Good, in this case, don't speculate about glycosylation please. Alternatively – confirm please by the suggested way.

In summary: the manuscript is still too speculative in the side of protein chemistry. The authors are invited to prove their statements in a proper way, or remove the statements (or appropriately modify them). Otherwise, I would suggest rejecting this manuscript, as conflicting with the journal standards.

REVIEWERS' COMMENTS

Reviewer #2 (Remarks to the Author):

In this revision, the authors have address most of the previous concerns. The most interesting new information is the striking decrease of SR45 and SAP18 protein levels in the double mutants. This make a stronger argument for an impact of ASAP/PASAP in the double mutants at the tested developmental stage. However, molecular interactions are temporally and spatially regulated in a dynamic fashion. A direct comparison between a vegetative stage setup in this manuscript with the results from inflorescence may or may not reflect the actual molecular dynamics regardless of developmental control. For example, SR45 is expressed at a much lower level in vegetative tissues than inflorescence (www.plantphysiol.org/cgi/doi/10.1104/pp.109.138180). It would not be a surprise if SAP18 exhibited a similar expression pattern. It is unclear if ACINUS and PININ would follow a similar expression pattern since the authors used overexpression lines in the experiment. Although it seems not directly relevant, it could be a reason for the difference in SAP18 protein level between this work and the report in inflorescence before if simply considering stoichiometry.

Response: We agree that the molecular interactions are dynamic and can be different in different tissues and cell types. We have added a phase to point out that the studies used different tissues : “in the inflorescence tissue”.

One correction that the authors did not make is the error in mutant line info. SALK_078854 is an insertion in gene AT3G25750, not ACINUS. The T-DNA line name for pinin mutant can not be found in the ABRC database either. It is an obvious concern if a paper publishes misleading information on the genetic materials it used.

Response: We appreciate this correction of our error. We have corrected the salk line to “SALK_078554”. The T-DNA line for pinin (*GABI_029C11*) is listed at the website: <http://signal.salk.edu/cgi-bin/tdnaexpress> and at ABRC as stock CS402723.

Reviewer #3 (Remarks to the Author):

The authors have conducted a significant amount of additional work in this revision that improves the manuscript and addresses the concerns raised by this reviewer. Minor editing is still required but should be easy to solve.

Reviewer #4 (Remarks to the Author):

after reading the answers of the authors and the revised manuscript, I have the following remarks.

1. It is very nice that the authors their experiment convincing. Yes, they good a good job, but it needs to be understood that IP-MS is not really state of the art. Therefore, two remarks need to be done: 1) the authors need to mark clearly that unknown number of interaction partners could be lost during handling; 2) some of 45 proteins representing the interactome could be precipitated via secondary, not native, interactions. Without these remarks the manuscript should not be accepted. Or – use doubt-free techniques.

Response: 1) We have added “However, some proteins, such as SPY and SEC, may interact transiently and were not detected by IP-MS.” on page 14. 2) We presume “secondary, not native, interactions” means direct not indirect interactions. We have added on page 14: “While our proteomic data does not distinguish the proteins that directly interact with AtACINUS from those that associate indirectly as a subunit of the interacting protein complexes, the greatly reduced levels of SR45 and AtSAP18 proteins in *acinus pinin* are consistent with the direct interactions predicted based on the conserved RSB domain.”

Whether IP-MS is state of the art is a subjective judgement which we will recuse from commenting. We believe IP-MS is a powerful (and popular) method for studying protein-protein interactions. Like any research method, it only works when it is done right with full consideration of limitations, which all research methods have. We hope the readers will appreciate the rigor we exercised in our use of IP-MS, considering the limitations of the method, compared to the standard practices. To obtain high-confident data, we performed two IP-MS experiments including total 5 replicates; we used native anti-AtACINUS antibody with *acinus* mutant as a control and used transgenic plants expressing AtACINUS-GFP with TAP-GFP as a control; we use both label-free MS and metabolic stable isotope labelling MS to quantify the differences between the sample and controls. We have not seen a more rigorous IP-MS study.

2. About the answer to my critique point # 4: the authors write: “As discussed above, the effects of spy mutation on the splicing of a large portion (4 of 10) of AtACINUS/PININ-dependent introns provide strong genetic evidence supporting the function of O-glycosylation of AtACINUS.”. Good, in this case, don’t speculate about glycosylation please. Alternatively – confirm please by the suggested way.

Response: The observation that loss of the glycosyl transferase (spy) glycosylating AtACINUS affects the (splicing) function of AtACINUS is genetic evidence supporting that glycosylation plays a role in regulating AtACINUS. We believe that it is important to point out appropriately the links between data/evidence and hypothesis, even though the evidence is not conclusive or definitive.

In summary: the manuscript is still too speculative in the side of protein chemistry. The authors are invited to prove their statements in a proper way, or remove the statements (or appropriately modify them). Otherwise, I would suggest rejecting this manuscript, as conflicting with the journal standards.

Response: We have made the modifications based on the comments that we can understand and agree with. This reviewer seems to have a unique view on our work or the journal standards that is not shared by the other reviewers. We appreciate the editor's ability to disagree with reviewers' suggestions on the decision.

** See Nature Research's author and referees' website at www.nature.com/authors for information about policies, services and author benefits